



# On etching, selection and measurement of confined fission tracks in apatite

Raymond Jonckheere[1*]

[1]  Geological Institute, Technical University and Mining Academy Freiberg, 09599 Freiberg, Germany

**Abstract**

This work investigates the selection of horizontal confined tracks for fission-track modelling. It is
carried out on prism sections of Durango apatite containing induced tracks with mean lengths of
~16, ~14, ~12, and ~10 μm. Suitable tracks are identified during systematic scans in transmitted
light. The explicit selection criteria are that the tracks are horizontal and measurable. We meas-
ure the length, width, orientation, and cone angle of each selected track and in some cases other
dimensions.
The confined track selection is in the first place dependent on a threshold width and in the second
place on the requirement that the tracks are etched to their ends. In most cases the first condition
implies the second, which decreases in importance as the tracks are shortened following anneal-
ing. The widest confined tracks, which must also be the shallowest, come to intersect the surface
and are excluded. In general, the selection is dominated by the width of the etched tracks. This, in
turn, depends on their orientation relative to the *c*-axis and the apatite etch rates, and their effec-
tive etch times. Despite the different geometrical configuration of the unetched host tracks and con-
fined tracks, neither the angular distribution nor the etch time distribution of the confined track
sample depends on the degree of annealing. This illustrates the general principle that those tracks
are selected that have the right properties for being selected. In this case etching-related factors
determining the track width are the most important, while the known geometrical biases are sec-
ond order. The track etch rate exhibits no demonstrable variation along the track, but significant
differences from track to track. Moreover, although the track etch rate of induced tracks is not
correlated with the extent of partial annealing, it is on average twice as high as the value for fossil
tracks.
Our length measurements are in good agreement with the annealing models for this apatite and
etch protocol. We submit that this is not fortuitous and that it is possible to select a representative
confined track sample, and perform reproducible and meaningful confined track length measure-
ments. Deliberate or inadvertent biasing, carelessness or inexperience will of course give differ-
ent results, but these should be treated as statistical outliers, not as an indication that track
lengths are fluid.
**Keywords:** apatite, fission-track, confined track etching, selection and measurement, track etch
rate

* Corresponding author; Raymond.Jonckheere@geo.tu-freiberg.de



## 1. Introduction

Fission-track analysis is a method for determining the ages of rocks and retracing their thermal
histories. It is based on the trails of lattice damage left in suitable minerals by the fragments of
fissioned uranium nuclei. The tools for interpreting fission-track data have evolved apace but
fundamental questions have remained unanswered. An important problem has to do with the
relationship between the actual damage trails and the fission tracks that are counted and meas-
ured after etching the mineral grains. Galbraith (2005) observed that "Inferring thermal histo-
ries from track measurements involves two steps. The first is to relate measurements made on
a sample to the true length distribution $f(l)$, and the second is to relate $f(l)$ to the thermal his-
tory". Several studies have addressed observational biases connected with measurements of
(horizontal) confined tracks. These are of a geometrical nature and their numerical treatment
is based on the so-called line-segment model. They include length bias, orientation bias, inter-
section (fracture and host track thickness) bias, and edge and surface proximity biases (Laslett
et al., 1982; 1984; Galbraith et al., 1990; Galbraith, 2002; 2005; Ketcham, 2003; 2005; Ketcham
et al., 2007).
In contrast to the geometrical biases, biases related to the actual etching of confined tracks are
less well understood. It was soon clear that the etching conditions (etchant, concentration, du-
ration and temperature) affected the lengths of confined tracks in apatite. Several experiments
with different etching protocols, listed in Jonckheere et al. (2017), showed that the mean length
of confined tracks first increased at a rapid rate up to a point where the tracks were considered
fully etched, following which it further increased at the much slower apatite etch rate (Laslett
et al., 1984). Watt et al. (1984) observed that the greater mean length of TinCLE's compared to
that of TinT's in the same sample could be explained by the time required for the host track to
etch, as compared to the almost immediate penetration of the etchant through a crack to reveal
the TinCLE's. Ketcham (2003) found that geometrical biases alone cannot account for the prop-
erties of confined track samples and proposed that "under-etching bias" was also an important
selection criterion.
Laslett et al. (1984) drew attention to the factors causing individual confined tracks to be etched
for a different time: "most confined tracks are over-etched in order to ensure that the etchant can
percolate down fractures and other tracks to etch all intersecting tracks (access time). Also, the
shortest tracks must be over-etched in order to ensure that the longest ones are fully revealed
(length)". These factors were integrated in numerical models (Rebetez et al., 1988; Ketcham
and Tamer, 2021), which showed that the lengths of etched confined tracks span the range from
zero length to that of the longest track. This implies that the measured track length distribution
is to some extent the product of an active selection based on a geometrical criterion or operator
bias (Ketcham and Tamer, 2021). Tamer et al. (2019) demonstrated the practical importance
of track selection; two operators mutually rejected ~14% of each other's selections. Ketcham
and Tamer (2021) guardedly concluded that the importance of selection "affects the fidelity of
thermal history modelling".
Step-etching of individual confined tracks in apatite showed that their lengths increase in erratic
fashion, with single 10s increments between ~0 and >1 µm, decreasing overall as the etch time
increases (5.5 M $HNO_3$ at 21 °C; Jonckheere et al., 2017). This underscores the importance of the
etch times of individual tracks for modelling of the confined track length distribution. Aslanian et
al. (2021) and Jonckheere et al. (2022) proposed a new etch model and measured the rates for
Durango apatite etched in 5.5 M $HNO_3$ at 21 °C. This permits to calculate the true duration ($t_E$) for
which an individual confined track has been etched from its orientation, shape and thickness. On
the one hand, it offers a practical criterion for selecting tracks for thermal history modelling, that
is independent of a person's judgement, and independent of the factors that caused a specific
measured track to have been etched for a given length of time. On the other, it presents a tool for
investigating the factors controlling $t_E$ or the $t_E$-distribution of the measured population. In the
next sections we investigate the influences of the track density and length using Durango apatites



annealed to nominal mean lengths of ~16, ~14, ~12 and ~10 μm (Ketcham et al., 2015; Aslanian
et al., 2022).

## 2. Materials and Methods

To investigate the effects of partial annealing on the composition and properties of the confined
track population we carried out measurements on four prism sections of Durango apatite dis-
tributed for inter-laboratory comparisons. The samples had been pre-annealed, neutron-irra-
diated and - all except one - annealed again to create induced-track populations with nominal
mean lengths of ~16 μm (sample 21-2), ~14 μm (sample 21-4), ~12 μm (sample 21-1) and ~10
μm (sample 21-3). Details of the irradiation and annealing conditions are given in Ketcham et
al. (2015). The track densities measured in transmitted and reflected light are reported in
Aslanian et al. (2021).
The samples were mounted in resin, ground on SiC papers, polished with 6-, 3-, and 1-μm dia-
mond suspensions, and given a final polish with 0.04-μm silica suspension, until the surface
was free of scratches under reflected light, and only faint scratches reappeared after etching.
All samples were etched for 20 s in 5.5 M $HNO_3$ at 21 C° (Carlson et al., 1999). The confined
track imaging was carried out with a motorized Zeiss AxioImager Z2m microscope connected
to a desktop computer running the Autoscan program. The samples were systematically scanned
in transmitted light at an optical magnification of 250× (100× Epiplan Neofluar dry objective
and 2.5× optovar). We made image stacks of each confined track considered suitable for meas-
urement. This meant that both its ends were free, distinct and well etched, and that a sharp
image of the entire track was contained within a single stack of six frames with a fixed spacing
of 0.25 μm. For an 8-μm track this corresponds to a dip angle of arcsin(1.25/8) ≈ 9°and an error
of <2%. The advantage of this criterion is that it is clear and amenable to mathematical descrip-
tion. The disadvantage is that it introduces a bias in favour of short tracks proportional to their
reciprocal length, which acts counter to the conventional length bias (Laslett et al., 1982; 1984;
Galbraith et al., 1990).
Depending on the case, we extracted the best image from a stack or compressed a part of it to a
single image, converted it to eight-bit and loaded it in CorelDraw, cutting out a square frame con-
taining a single track to add to the database (**Figure 1a-d**). For calculating its effective etch time
($t_E$) we measured its maximum width ($r_0$) at the intersection with its host track (TinT; **Figure
1e**) or, in rare cases, with a crack (TinCLE). For calculating the track etch rate ($v_T$), we measured
a second width ($r_1$) at some distance ($s_1$) from the first along a straight section of track. This
works for tracks at most angles to the **c**-axis ($\phi \lesssim 80°$) but not for those at the highest ($\phi \gtrsim$
80°), where part of the channel is obscured by a diamond-shaped etch figure (**Figure 1f**; Jonck-
heere et al., 2022). As this figure is bounded by the fastest etching faces, a precise $t_E$-estimate is
obtained from measurements of the distances between opposing sides ($d_1$ and $d_2$). A less precise
lower estimate can still be obtained from the width of the channel outside the diamond shape
($r_0$).
Some tracks at ≳45° to the **c**-axis in the most annealed sample have an atypical shape due to
"interrupted" etching at a "gap" in the latent track ("unetchable gap", Green et al., 1986). We
distinguish between "stepped" tracks where the gap was pierced during etching (**Figure 1c**)
and "gapped" tracks where it was not (**Figure 1d**; Galbraith, 2005). The latter are characterized
by their short lengths and irregular endings. We divided stepped tracks in three segments for
measurement (**Figure 1g**). The first, from the host track intersection to the gap, has length $l_1$;
the second, from the intersection in the opposite direction, has length $l_2$; the third, from the gap
to the other end, has length $l_3$. The maximum width at the intersection with the host track is $r_0$;
those on either side of the gap are $r_1$ and $r_3$; the widths along segments 2 and 3 at a distance
from $r_0$ and $r_3$ are $r_2$ and $r_4$.
From these measurements, we calculated the effective etch time ($t_E$) of each confined track as:

$$t_E(s) = 30 \ \frac{r_0(\mu m)}{v_R(\mu m/min)} \qquad\qquad (\phi \lesssim 80°) \qquad\qquad\qquad (1a)$$

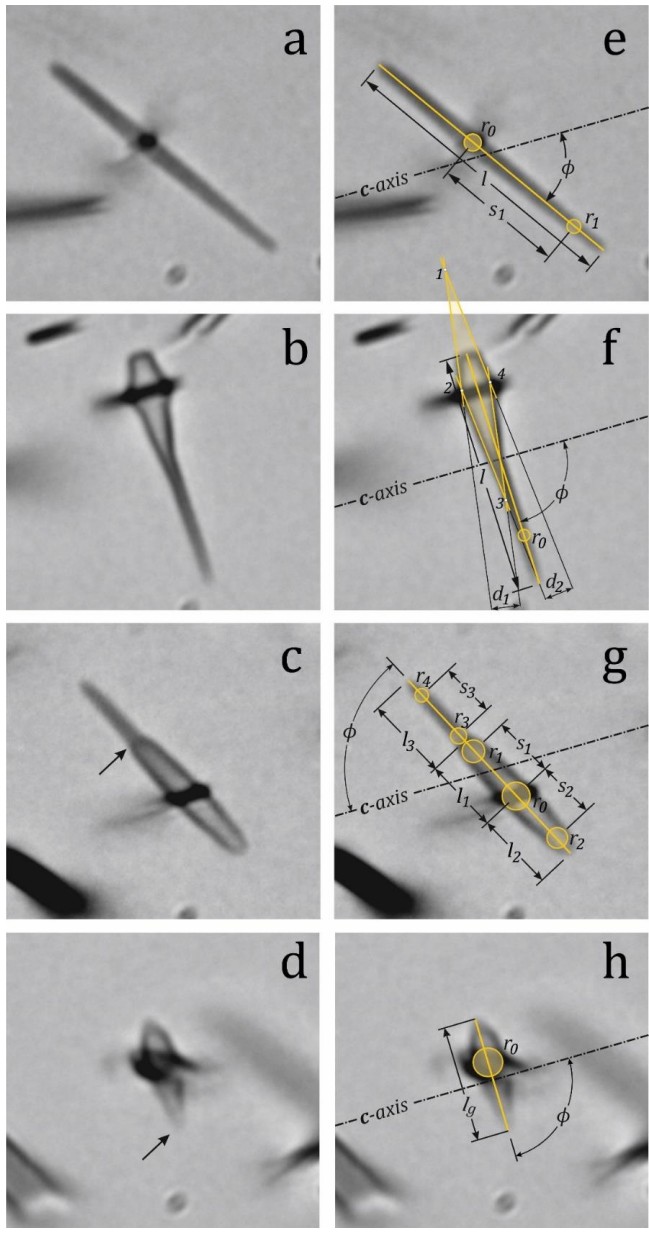

**Figure 1**. Etched confined tracks distinguished in this work: (**a**) continuous track at a low to moderate angle to the $c$-axis; (**b**) continuous track at a high angle to the $c$-axis, with a diamond shaped etch figure at its intersection with the host track; (**c**) stepped track showing narrowing (arrow) due an interruption of along-track etching; (**d**) gapped track with an abnormal termination (arrow); (**e**) length ($l$), $c$-axis angle ($\phi$) and width ($r_0$) of track (a); the width at $r_1$ and its distance ($s_1$) to $r_0$ are used to calculate the cone angle ($\theta$) and the etch rate ($v_T$) of the track (equations 3a and b); (**f**) measurement of the distances ($d_1$ and $d_2$) between facing sides of the diamond shape at track (b), bounded by the fastest etching apatite faces, used for calculating its effective etch time ($t_E$; equation 1b); (**g**) measurements of widths ($r_0$, $r_1$, $r_2$, $r_3$, and $r_4$) of track (c) aimed at determining its effective etch time ($t_E$), the etch delay ($t_D$) at the constriction and the etch rates of the different track sections; (**h**) measurement of the length ($l$), width ($r_0$) and orientation ($\phi$) of the gapped track in (d).



$$t_E(s) = 15 \frac{(d_1 + d_2)(\mu m)}{V_{R,MAX}(\mu m/min)} \qquad (\phi \gtrsim 80°) \tag{1b}$$

wherein $v_R$ is calculated from the model of Aslanian et al. (2021), with $\phi_W = 90 - \phi$, and $v_{R,MAX} \approx$
3.0 µm/min:

$$v_R(\mu m/min) = -0.0071\,\phi_W^2 + 0.2807\,\phi_W + 0.2495 \qquad (\phi_W \lesssim 20°) \tag{2a}$$

$$v_R(\mu m/min) = 0.00025\,\phi_W^2 - 0.0633\,\phi_W + 4.2500 \qquad (\phi_W \gtrsim 20°) \tag{2b}$$

The track etch rate $v_T$ is calculated as:

$$v_T(\mu m/min) = \frac{v_R(\mu m/min)}{\sin(\theta/2)} \tag{3a}$$

wherein $\theta$ is the angle between facing straight margins of the track, calculated from $r_0$, $r_1$ and $s_1$
as:

$$\theta = 2\arcsin\left(\frac{(r_0 - r_1)(\mu m)}{2\,s_1(\mu m)}\right) \tag{3b}$$

The expected minimum and maximum widths for each track orientation are calculated from $v_R$
(eqs. 2a and 2b) as:

$$r_{MAX}(\phi)\,(\mu m) = \left(\frac{2}{3}\right)(min)\,v_R(\phi_w)\,(\mu m/min) \tag{4a}$$

$$r_{MIN}(\phi)\,(\mu m) = \left(\frac{1}{5}\right)(min)\,v_R(\phi_w)\,(\mu m/min) \tag{4b}$$

where $r_{MAX}$ refers to tracks etched for the full 20 s immersion time $t_I$, and $r_{MIN}$ to the minimum
time required for a track to be etched from its midpoint to both ends ($\approx 7.5(\mu m)/75(\mu m/min)$;
Aslanian et al., 2021). We also estimated the maximum ($t_{E,MAX}$) and minimum ($t_{E,MIN}$) effective
etch times. The maximum assumes that a track etches from the moment that the sample is im-
mersed in the etchant till it is taken out and rinsed. The minimum assumes that a track must be
etched for long enough to reach a width of ~0.3 µm, in order to be selected for measurement
(Aslanian et al., 2021)

$$t_{E,MAX}(\phi)\,(s) = t_I \tag{5a}$$

$$t_{E,MIN}(\phi)\,(s) = 30\,\frac{0.3\,(\mu m)}{v_R(\phi_w)\,(\mu m/min)} \tag{5b}$$

## 3. Results and Discussion

From 2170 track images taken by one participant the other rejected 20 as not suitable for meas-
urement. This <1% rejection rate is much lower than the averages of both participants in the
investigation of Tamer et al. (2019; ~14%). However, the present is a one-way rate concerning a
single set of images taken using one set of etching and observation conditions. Also in contrast to
the present, the latter investigation was of in general lower densities of fossil tracks in [252]Cf-irra-
diated samples.

### 3.1 Track lengths

Plots of the measured ($l$) and the *c*-axis projected lengths ($l_P$; Donelick et al., 1999) of horizontal
confined tracks against angles to the *c*-axis ($\phi$) illustrate the known length shortening and in-
creasing anisotropy with increasing annealing (**Figure 2**; Green et al., 1986; Donelick, 1991). The
mean lengths ($l_M$) are close to the values predicted by the annealing equations for Durango ap-





atite etched 20 s in 5.5 M $HNO_3$ at 21 °C (Ketcham et al., 1999; **Table 1**). The maximum differ-
ence (0.15 µm) is that between the measured mean length (10.45 µm) and that predicted by
the fanning rectilinear model for the most annealed sample (21-3; 10.30 µm). All other differ-
ences between measured and predicted mean lengths are <0.05 µm. The standard deviations
of the length distributions ($s_M$) are also in agreement with model predictions, with a maximum
difference between the measurements and models of 0.08 µm for sample 21-1 (**Table 1**). The
relationship between the standard deviations ($s_M$) and means ($l_M$) of the track length distributions
as well as that between the **c**-axis ($l_C$) and **a**-axis intercepts ($l_A$) of ellipses fitted to the length *vs.*
orientation data are consistent with the equations of Donelick et al. (1999; **Figure 3a**). The agree-
ment between our data and those reported in Carlson et al. (1999) shows that independent sci-
entists working two decades apart on samples annealed at different conditions, using different
equipment and measurement methods, but the same etching protocol, nevertheless produce con-
sistent results. We believe that this is not without significance and return to the issue later in the
discussion.
**Figure 2** plots the **c**-axis-projected length of each track against its actual angle to the axis (Donelick
et al., 1999). The agreement between the mean **c**-axis-projected lengths ($l_P$) and the annealing
models (Ketcham et al., 1999) is somewhat worse than for the non-projected lengths. The calcu-
lated values are 0.21-0.33 µm above the predictions of both the rectilinear and curvilinear models
(Table 1). In contrast, the standard deviations ($s_P$) are all within 0.05 µm of their predicted values.
The relationship between $s_P$ and $l_P$ is again consistent with the equation of Donelick et al. (1999;
**Figure 3b**). Except for a small offset of 0.2-0.5 µm, the **c**-axis ($l_{PC}$) and **a**-axis ($l_{PA}$) intercepts of
regression lines fitted to the ($l_P$, $\phi$)-data are almost identical, indicating that **c**-axis projection ef-
fectively eliminates the anisotropy of the track lengths in the annealed and unannealed samples
(**Figure 3a**).
Although it is most noticeable in the case of sample 21-3 (**Figure 2h**), $l_P$ is more tightly distributed
about the local mean at greater angles to the **c**-axis in all four samples. This is not related to the
measurements but to the **c**-axis projection, which funnels tracks at higher angles to the **c**-axis into
a narrower length interval than those at lower angles (Donelick et al., 1999). The **c**-axis projection
thus trades the unequal means $l_{PM}$ of the length distributions at different **c**-axis angles for unequal
standard deviations $s_{PM}$. The effect is quite pronounced; the standard deviations of all confined
tracks lengths at <15° to the **c**-axis are on average twice as high as those of tracks at >75° to the
**c**-axis (**Figures 3b and B1** of Appendix B). In the case of sample 21-3, $s_{PM}$ decreases from ~0.8
µm at ~0° to the **c**-axis to ~0.2 µm at ~90°. It is not clear how this affects thermal histories mod-
elled using **c**-axis projection, but it would appear that careful accounting for the orientations of
the measured tracks is essential to avoid artefacts.
**Figure B2** of appendix B shows the length histograms of tracks at 15° angular intervals, projected
onto the **c**-axis while preserving length differences, i.e., keeping the distance from each data point
to the fitted ellipse fixed. This eliminates length anisotropy without affecting the distributions
about the means. However, because it applies to tracks annealed under identical conditions and
not to variable-temperature histories it is of no use for modelling. It is nevertheless interesting to
contrast both projections. The latter understands each length measurement as an estimate of the
mean length in a given orientation but departing from it due to accidents of track formation and
etching, but not due to annealing. The former interprets each single length measurement as the
mean of a population, whose value is a direct reflection of its annealing history, excluding random
effects. As we understand it, the former is assumption more correct but the latter a condition for
modelling.
However the assumption implicit in the common **c**-axis projection has interesting implications.
In contrast to individual track lengths, two mean track lengths projected onto the **c**-axis differ due
to their thermal histories, not due to accidents of track formation or etching, which do not affect
the means. Since annealing unidirectionally lowers the mean lengths, a population with a shorter



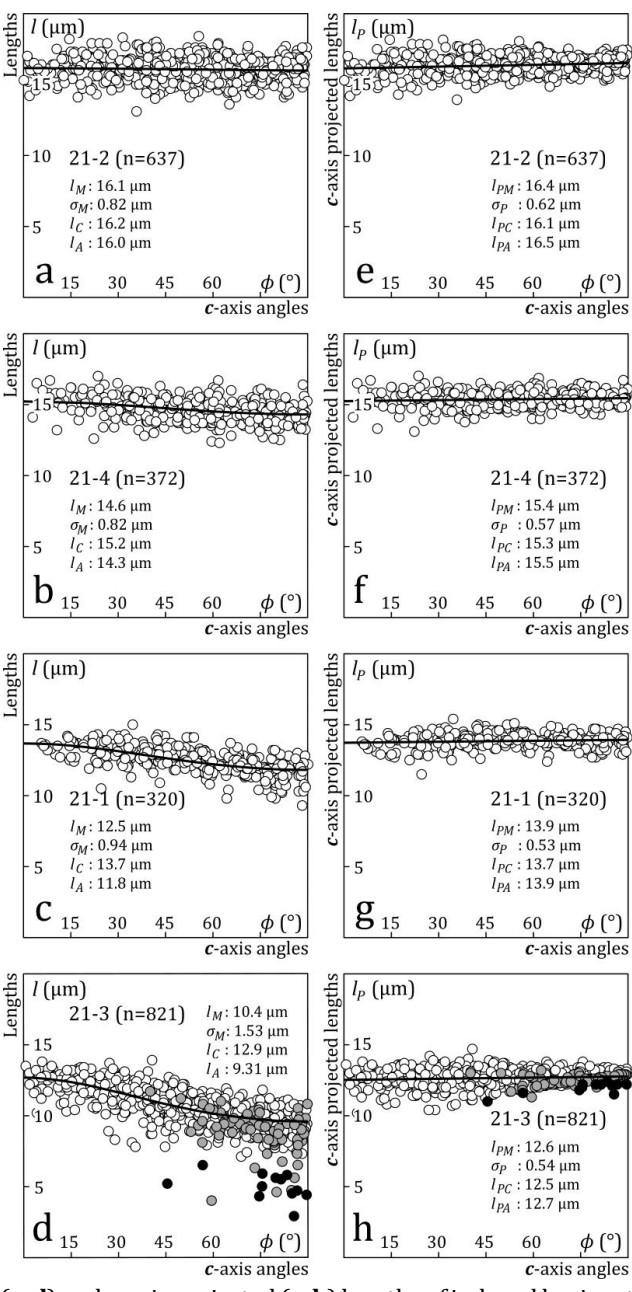

**Figure 2.** Measured (**a-d**) and *c*-axis projected (**e-h**) lengths of induced horizontal confined tracks in the four studied samples plotted against angle to the *c*-axis: (**a**, **e**): unannealed; (**b**, **f**): annealed 10 h at 240 °C; (**c**, **g**): annealed 10 h at 288 °C and (**d**, **h**) annealed 10 h at 310 °C (Ketcham et al., 2015). White circles: continuous tracks; grey circles: stepped tracks; black circles: gapped tracks. The solid lines in **a-d** are ellipses fitted to the data (excluding data for stepped and gapped tracks in **d**); those in **e-h** are linear regression lines; $n$: number of measured lengths; $l_M$: mean track length; $s_M$: standard deviation of the length distribution; $l_C$: *c*-axis intercept of the fitted ellipse; $l_A$: *a*-axis intercept of the ellipse; $l_{PM}$: mean *c*-axis projected length; $s_{PM}$: standard deviation of the *c*-axis projected lengths; $l_{PC}$: *c*-axis intercept of the fitted regression line; $l_{PA}$: *a*-axis intercept of the regression line.

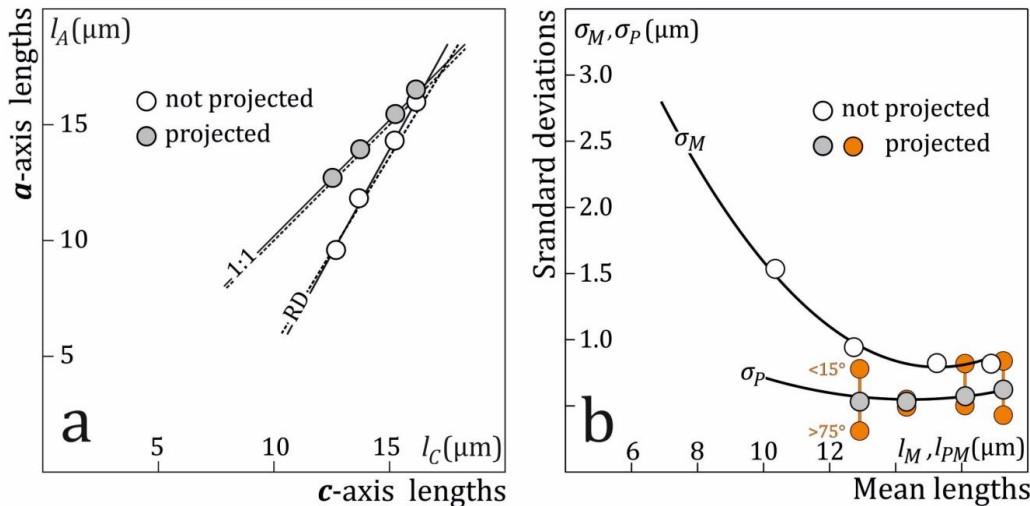

**Figure 3**. (**a**) Relationship between the *c*-axis and *a*-axis intercepts of ellipses fitted to the measured lengths (white circles) and regression lines fitted to the *c*-axis projected lengths (grey circles) of confined tracks in the studies samples; solid lines represent published equations (RD: Donelick et al., 1999; 1:1: isotropic trend), dashed lines have been fitted to the measurements in this work; (**b**) relationship between the standard deviations and means of the distributions of the measured (white circles) and *c*-axis projected lengths (grey circles) of horizontal confined tracks in the studies samples; the orange circles represent the standard deviations of tracks at <15° and >75° to the apatite *c*-axis.





| Sample | | $\phi$(°) | $l$ (μm) | $l_P$ (μm) | $r_0$ (μm) | $r_1$ (μm) | $s_1$ (μm) | $\theta$ (°) | $d$ (μm) | $t_E$ (s) | $v_T$ (μm/min) |
|---|---|---|---|---|---|---|---|---|---|---|---|
| 21-2 | Count | 637 | 637 | 637 | 629 | 553 | 553 | 500 | 139 | 712 | 500 |
| | Mean | 53.1 | 16.1 | 16.4 | 0.68 | 0.53 | 7.56 | 2.92 | 1.00 | 11.0 | 208 |
| | Error | 0.89 | 0.03 | 0.02 | 0.01 | 0.01 | 0.10 | 0.07 | 0.02 | 0.12 | 22 |
| | S.Dev | 22.5 | 0.82 | 0.62 | 0.23 | 0.18 | 2.33 | 1.63 | 0.21 | 3.22 | 118 |
| 21-4 | Count | 374 | 372 | 372 | 337 | 337 | 337 | 335 | 49 | 366 | 335 |
| | Mean | 54.2 | 14.6 | 15.4 | 0.79 | 0.63 | 6.04 | 3.53 | 1.05 | 11.9 | 165 |
| | Error | 1.10 | 0.04 | 0.03 | 0.01 | 0.01 | 0.16 | 0.10 | 0.03 | 0.17 | 18 |
| | S.Dev | 21.2 | 0.82 | 0.57 | 0.22 | 0.19 | 2.86 | 1.80 | 0.21 | 3.26 | 84 |
| 21-1 | Count | 320 | 320 | 320 | 308 | 281 | 281 | 213 | 66 | 249 | 209 |
| | Mean | 53.5 | 12.5 | 13.9 | 0.76 | 0.63 | 5.96 | 3.14 | 1.33 | 12.0 | 201 |
| | Error | 1.23 | 0.05 | 0.03 | 0.02 | 0.01 | 0.11 | 0.13 | 0.03 | 0.21 | 14.0 |
| | S.Dev | 21.9 | 0.94 | 0.53 | 0.27 | 0.23 | 1.87 | 1.89 | 0.28 | 3.37 | 115 |
| 21-3 | Count | 821 | 821 | 821 | 724 | 713 | 713 | 684 | 165 | 826 | 684 |
| | Mean | 53.2 | 10.4 | 12.6 | 0.67 | 0.52 | 4.93 | 4.03 | 1.00 | 10.6 | 159 |
| | Error | 0.77 | 0.05 | 0.02 | 0.01 | 0.01 | 0.07 | 0.11 | 0.02 | 0.12 | 26 |
| | S.Dev | 22.1 | 1.53 | 0.54 | 0.20 | 0.16 | 1.92 | 2.94 | 0.23 | 3.37 | 91 |

**Table 1A.** Average *c*-axis angles, lengths and widths of horizontal confined induced tracks in prism faces of Durango apatite and calculated track effective etch times $t_E$ and track etch rates $v_T$. Etching conditions: 20 s in 5.5 M HNO$_3$ at 21 °C; the different measured lengths and widths are shown in Figure 1.

| Sample | | $\phi$(°) | $l$ (μm) | $l_P$ (μm) | $r_0$ (μm) | $r_1$ (μm) | $s_1$ (μm) | $\theta$ (°) | $d$ (μm) | $t_E$ (s) | $v_T$ (μm/min) |
|---|---|---|---|---|---|---|---|---|---|---|---|
| Continuous | Count | 731 | 731 | 732 | 668 | 658 | 658 | 629 | 103 | 735 | 629 |
| | Mean | 50.6 | 10.7 | 12.7 | 0.66 | 0.52 | 4.86 | 3.56 | 0.97 | 10.7 | 82 |
| | Error | 0.80 | 0.05 | 0.02 | 0.01 | 0.01 | 0.07 | 0.08 | 0.02 | 0.13 | 1.9 |
| | S.Dev | 21.7 | 1.26 | 0.54 | 0.20 | 0.16 | 1.92 | 2.00 | 0.25 | 3.48 | 48 |
| Stepped | Count | 78 | 78 | 61 | 50 | 49 | 49 | 49 | 52 | 79 | 49 |
| | Mean | 73.9 | 8.93 | 12.6 | 0.89 | 0.42 | 5.90 | 10.1 | 1.05 | 10.3 | 39 |
| | Error | 1.31 | 0.17 | 0.04 | 0.02 | 0.02 | 0.21 | 0.77 | 0.03 | 0.23 | 3.0 |
| | S.Dev | 11.6 | 1.50 | 0.31 | 0.17 | 0.14 | 1.44 | 5.41 | 0.18 | 2.04 | 21 |
| Gapped | Count | 12 | 12 | 12 | 6 | 6 | 6 | 6 | 10 | 12 | 6 |
| | Mean | 76.6 | 5.02 | 11.9 | 0.58 | 0.49 | 4.14 | 3.51 | 1.00 | 8.52 | 152 |
| | Error | 3.76 | 0.27 | 0.11 | 0.08 | 0.08 | 1.30 | 1.09 | 0.07 | 0.82 | 49 |
| | S.Dev | 13.0 | 0.95 | 0.39 | 0.20 | 0.19 | 3.20 | 2.67 | 0.21 | 2.84 | 119 |

**Table 1B.** Breakdown of the data for sample 21-3 in Table 1A according to the appearance of the etched track (Figure 1; continuous, stepped or gapped).





mean length has experienced more severe annealing than one with a longer mean length. Thus,
the order of projected lengths corresponds to the order of formation of the tracks. The number and
lengths of confined tracks thus divide a sample's history in a corresponding number of time-inter-
vals to each of which corresponds a specific mean length. This allows to convert its age and length
distribution directly to a Tt-path, without the need to search Tt-space. Allowance can be made for
biases, e.g., the fact that shorter tracks represent a larger population and time interval than longer
tracks. The uncertainties concerning the exact time of formation of each population can also be
taken into consideration (Jonckheere and Ratschbacher, 2010). The single solution represents
the core of the set of T,t-solutions consistent with the data and measurement uncertainties
(Rebetez et al., 1994).
Sample 21-3 is characterized by the occurrence of gapped and stepped tracks (**Figure 1**) at
>45° to the *c*-axis (**Figure 2d**). This is less than the calculated value for Durango apatite, both
for the predicted ($\phi_{alr}$ >60°; $\phi_{alr}$ = angle of accelerated length reduction; Donelick et al., 1999)
and the measured lengths ($\phi_{alr}$ >80°). The occurrence of stepped and gapped tracks increases
with increasing angle to the *c*-axis (**Figure 4a**). This correlates with a rapid drop of the apatite
etch rate along the track axis from <2 µm/min at 45° to ~0.5 µm/min perpendicular to the *c*-
axis (**Figure 4b**). This, together with the simultaneous appearance of gapped and stepped
tracks, argues for the formation of non-etchable gaps (Green et al., 1986) rather than for a more
general form of accelerated length reduction (Donelick et al., 1999), and therefore for a breakup
of the tracks and a discontinuous structure of latent tracks (Paul and Fitzgerald, 1992; Paul,
1993). It appears that *c*-axis projection converts the lengths of gapped and stepped tracks to
values within a narrow length interval that depends little on their original lengths and orienta-
tions. It thus seems that less precise measurements of these lengths or orientations are not a
great concern.
The gapped tracks in sample 21-3 have lengths <7 µm (**Figure 1h**; Green et al., 1986), which is
somewhat shorter that the main segments of the stepped tracks, intersected by the host tracks
(**Figure 1g**; $l_1 + l_2$). The total lengths of the stepped tracks, including the section after the gap
(**Figure 1g**; $l_3$), are on average ~0.2 µm shorter than continuous tracks with the same orienta-
tions (**Figure 4a**; $l_t$) due to the delay caused by the gap. The lengths of continuous tracks ($l_t$), the
final lengths of stepped tracks ($l_1 + l_2 + l_3$), that of their main sections ($l_1 + l_2$), and even the lengths
of gapped tracks ($l_g$) appear to exhibit a similar linear decrease with increasing angle to the *c*-
axis. In contrast, $l_3$ exhibits no visible dependence on orientation and no discernible preferred
length between zero and half the total track length (**Figure 4c**). This suggests that non-etchable
gaps can form more or less along the entire length of the tracks at this stage of annealing. Substi-
tuting $r_1$ and $r_3$ for $r_0$ in equation (1a) we calculated the difference of the effective etch time $\Delta t_{E1,3}$
on either side of the gap and interpreted it as the time required to pierce the gap at the etch rate
of undamaged apatite $v_R$ in the direction of the track in order to estimate the size of the gaps. The
results range from <10 nm to >100 nm, with nevertheless a pronounced mode at ~30 nm, which
is consistent with TEM observations on latent tracks (**Figure 4d**; Paul and Fitzgerald, 1992; Paul,
1993) (Table 3).

### 3.2 Track widths

**Figure 5** plots the widths of the confined track against their angles to the *c*-axis. The data for all
samples define a similar boomerang shape, with minima parallel and perpendicular to the *c*-axis,
and the greatest widths at 60-75°. The solid curve (1) defines the maximum achievable widths
after 20s immersion in 5.5 M $HNO_3$, calculated with the etch rates of Aslanian et al. (2021). Few
tracks attain that width due to the variable but finite access times required for the etchant to
travel (etch) down the host tracks and across to the confined tracks (Rebetez et al., 1988;
Ketcham and Tamer, 2021). For illustration, the shaded band under (1) represents access times
of up to 4 seconds. There also appears to be a distinct lower limit (2), which we interpret as the
minimum width for a track to be selected for measurement. Our lower limit at ~0.3 µm is lower
than the value of Aslanian et al. (2021; ~0.5 µm), perhaps due to their longer immersion time or

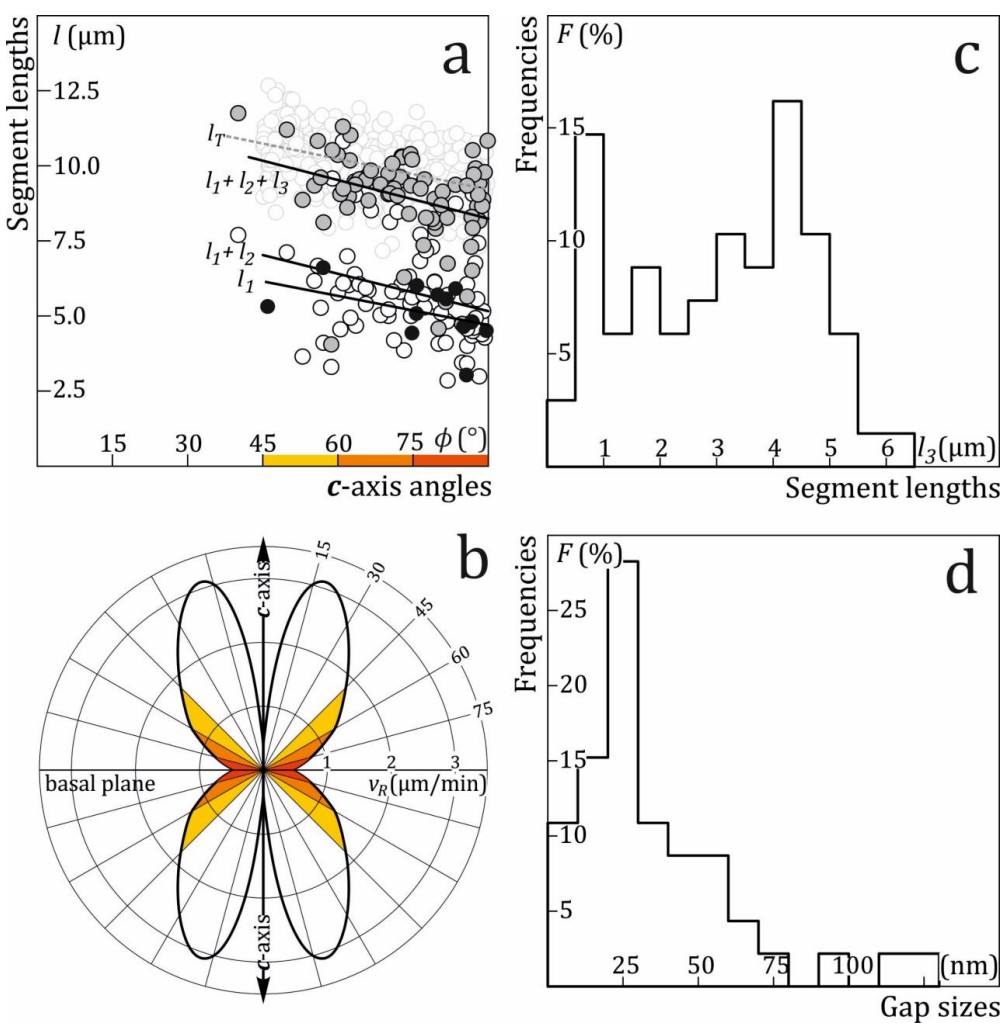

**Figure 4.** (**a**) Measurements of stepped and gapped tracks plotted against angle to the $c$-axis ($\phi$); white circles: lengths of the main sections of stepped tracks (**Figure 1g**; $l_1 + l_2$); grey: full lengths of stepped tracks ($l_1 + l_2 + l_3$); black: lengths of gapped tracks ($l_1$); grey-edged circles and dashed line show the full lengths of continuous tracks ($l_T$) in the same interval, for comparison; the colour-coding along the horizontal axis refers to angular intervals with decreasing apatite etch rates in the etch rate plot (b); (**b**) apatite etch rate plot from Aslanian et al. (2021), colour-coded to correspond with (a); (**c**) frequency distribution of the lengths of the distal segments of stepped tracks ($l_3$; past the constriction); (**d**) frequency distribution of the sizes of perforated gaps between the main and distal segments of stepped tracks, calculated from the difference of the widths at $r_1$ and $r_3$ (**Figure 1g**).

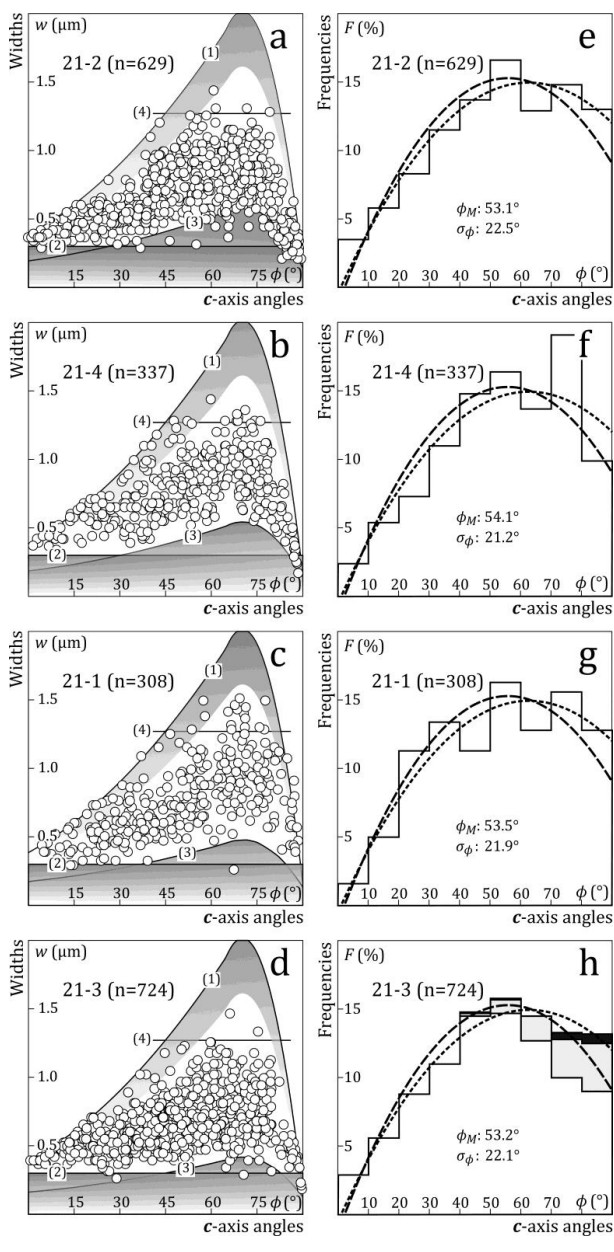

**Figure 5.** (**a-d**) Widths of horizontal confined tracks (**Figure 1e**; $r_0$; excluding high-angle tracks) plotted against their angles to the ***c***-axis ($\phi$), and the main factors that guided their selection; (1) maximum width of confined tracks that started etching between 0 and 4 seconds after immersion; (2) overall minimum width of tracks accepted for measurement unless condition (3) applies; (3) minimum width of a track etched to both ends, calculated using equation (6); (4) approximate limit above which tracks close to the surface become too wide to remain confined; (**e-h**) distributions of track angles to the ***c***-axis for the four studied samples; white: continuous tracks; grey: stepped tracks; black: gapped tracks; the short-dashed line is a polynomial fit to the combined data for the four samples (including high-angle tracks); the long-dashed line is a polynomial fit to the range of widths at each orientation delineated by constraints (1)-(4) (i.e., excluding high-angle tracks).





to different track selection criteria. Over a broad range of orientations (≲45° to ≳75°), the mini-
mum track widths are above this limit. This is due to the minimum time required for a track to
etch to its ends, so that it is suitable for measurement. For the sake of calculation, we assume that
both ends need to attain a width of ~0.2 μm for a track to be measurable. Assuming that a track
of length $l$ etches from the middle toward both ends, we find that $t_{E,MIN} = \frac{1}{2}$ ($l/v_T + 0.2/v_R$). The
corresponding minimum width is:

$$w_{MIN}\ (\mu m) = \left(\frac{l\ (\mu m)}{2}\right)\left(\frac{v_R\ (\mu m/min)}{v_T\ (\mu m/min)}\right) + 0.2\ (\mu m) \qquad (6)$$

This estimate depends on both the length and orientation of the tracks, and is shown as curve (3)
in **Figure 5**. Despite the fact that these limits are diffuse, i.e., dependent on variable access times
and intersection points, they appear to constrain the data rather well, at least in a qualitative
manner. Within these bounds, there is nevertheless a conspicuous lack of tracks wider than ~1.25
μm at 60-75° to the **c**-axis (4). To attain greater widths the access times must be short, which
means that the tracks must be close to the surface, which is itself lowered at a slow rate during
the immersion of the sample. We conclude that the lack of tracks wider than ~1.25 μm is due to
the fact that broad, shallow tracks come to intersect the surface as etching proceeds. This is a
form of surface proximity bias (Galbraith et al., 1990), although there is in our data no obvious
correlation with the track length.
The distance $\Delta w$ between the lower of the upper limits (1 and 4) and the higher of the lower limits
(2 and 3) is proportional to the range of track widths consistent with these four criteria at a given
orientation. **Figures 5e-h** plot the average $\Delta w$ for our four samples against angle to the **c**-axis,
ignoring the small differences between them. The result is a good fit to the combined and individ-
ual distributions of the track orientations. The agreement with the measured distributions is in
fact better than shown in **Figure 5e-h** since the latter include high-angle tracks not included in
**Figure 5a-d** because their effective etch times were calculated as in **Figure 1f**. This is an indica-
tion that etching- and observation-related factors have as least as much influence on the angular
frequencies of confined tracks as geometrical factors (Ketcham, 2003). The angular distributions
of our four samples are indistinguishable when the stepped and gapped tracks are included in
the measurements of the most annealed sample. This extends the range of constant angular bias
to within the domain of track break-up, i.e., somewhat beyond the limit assumed by Galbraith et
al. (1990) and Ketcham (2003).
Thus, in samples with low to moderate annealing, confined tracks are in the first place selected
based on their width, in the form of a minimum width independent of their length and orientation
(2) and of the rate of width increase (etch rate) in a given orientation, which determines the frac-
tion of tracks above the threshold (1). This selection is modified by factors depending on width
and length (3, 4), i.e., surface proximity bias (Galbraith et al., 1990) and the fact that longer tracks
on average attain a greater width before they are etched to their ends (Laslett et al., 1984). How-
ever, the influence of the track length on the angular frequencies of horizontal tracks is not no-
ticeable within the resolution of our measurements and the annealing range of our samples ($l_M$ =
16.1 -10.5 μm). It seems fitting to describe these as "(under-)etching biases" (Ketcham 2003). At
advanced stages of annealing, track selection comes to be controlled by "track loss" (Green et al.,
1986; Ketcham, 2003).

### 3.3 Effective etch time

**Figures 6a-d** plot the effective etch times $t_E$ of individual confined tracks against their angles to
the **c**-axis; $t_E$ is calculated from the widths in **Figure 5a-d** using equations (1a) and (2a-b) and the
apatite etch rates of Aslanian et al. (2021). The effective etch times of the high-angle tracks (φ
>80°), not included in **Figure 5a-d**, are calculated with equation (1b) and plotted with a different
symbol; the two estimates are consistent with each other. The selection boundaries (1-4) are con-
verted to effective etch time limits in the same manner. Together the data again define boomer-
ang shapes, although inverted compared to those in **Figure 5a-d**, with the thinnest tracks at low



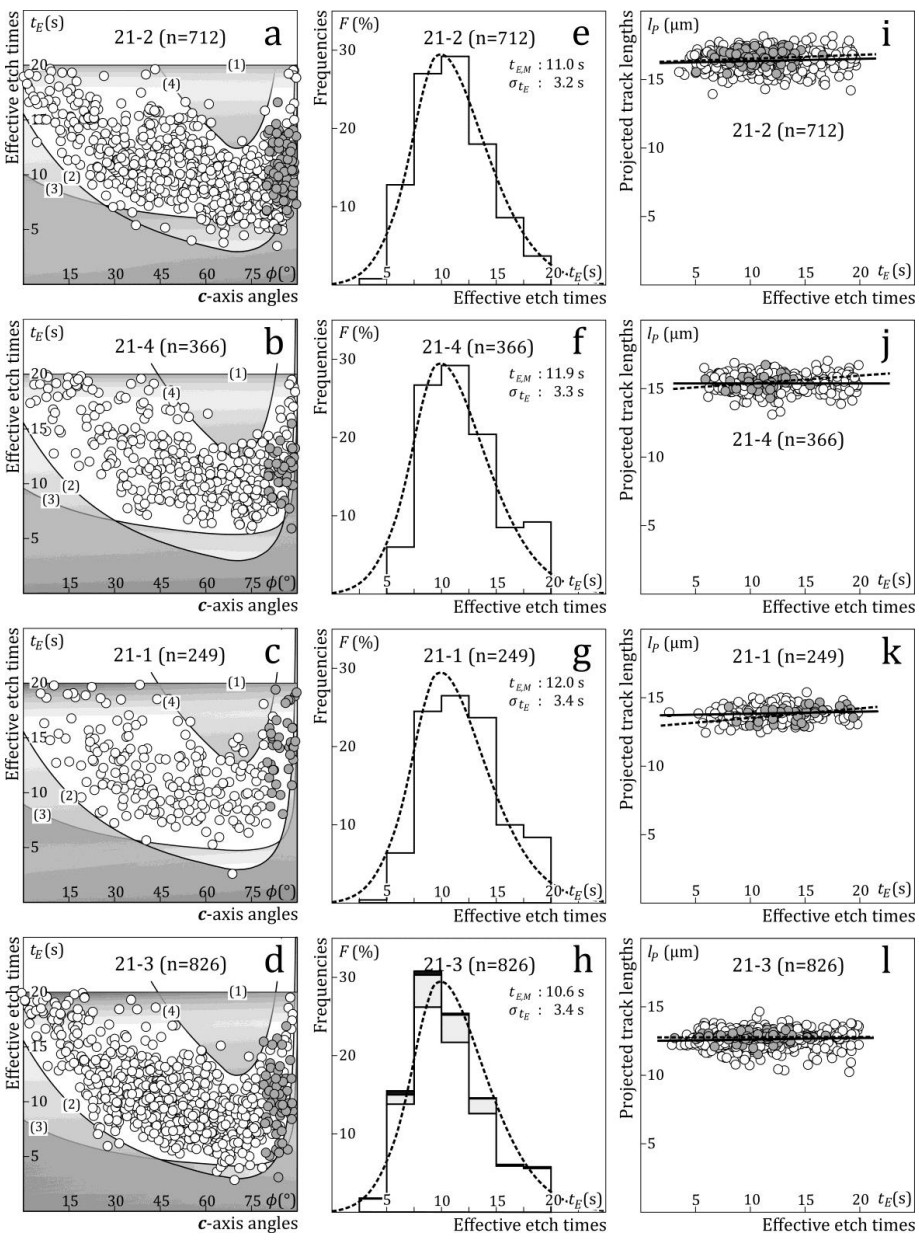

**Figure 6**. (**a-d**) Effective etch times of horizontal confined tracks ($t_E$) in the four studied samples plotted against their angles to the ***c***-axis ($\phi$), and the main factors, (1)-(4), that guided track selection, taken from **Figure 5**; white: low and moderate angle tracks (**Figure 1e**); grey: high-angle tracks (**Figure 1f**); (**e-h**) effective etch time distributions for the four studied samples; white: continuous tracks; grey: stepped tracks; black: gapped tracks; the short-dashed line is a polynomial fit to the combined data for the four samples; $t_{E,M}$ and $s_{t_E}$: arithmetic mean and standard deviation of the etch-time distribution (**i-l**) ***c***-axis projected lengths ($l_P$) of confined tracks plotted against their effective etch times ($t_E$); white: low and moderate angle tracks (**Figure 1e**); grey: high-angle tracks (**Figure 1f**); the solid black line is a linear regression line fitted to the low and moderate angle data (white); the dashed line is a regression line fitted to the high-angle data (grey).





and high angles to the *c*-axis having the longest etch times and the broadest tracks the shortest.
This illustrates the fact that those tracks are selected that have the right properties for being se-
lected and that the dominant selection criterion across all samples, lengths and orientations is
width. Longer low-angle tracks and shorter high-angle tracks widen at a comparable slow rate
(Aslanian et al., 2021) and are both measured after etch times close to the total immersion time,
while tracks in intermediate orientations, which widen at a faster rate, are selected after a shorter
time. High-angle tracks like that in **Figure 1b** are an exception because their effective etch time
is calculated from the diamond shaped figure at its intersection with the host track (Jonckheere
et al., 2022).
The etch time distributions are all right skewed, with similar means and standard deviations (**Fig-**
**ure 6e-h**). This might be unexpected, given the different track lengths and densities of the sam-
ples (Aslanian et al., 2022), which affect the number and length of the host tracks, their distances
to the nearest confined tracks, and therefore the access time and the etch time distributions. The
distributions in **Figure 6e-h** are however those of *selected* tracks, not of the sampled population
as such, and as long as it contains tracks meeting fixed selection criteria their distribution remains
the same. The etch time distribution is controlled by the threshold width (2) and the immersion
time (1). This does not exclude geometrical biases; apart from through length bias (Laslett et al.,
1982), track length influences the selection through constraints (3) and (4). However, their effect
on samples with simple length distributions, like those investigated here, appears to be small.
The operator influences track selection through the threshold width (2) and the assessment
whether tracks are etched to their ends (3). These decisions are not independent as a high thresh-
old width implies that most selected tracks are etched to their ends. E.g., **Figure 6a-d** suggests
that a threshold width of ~0.7 μm would ensure that all selected track lengths are suitable for
measurement. The actual value is somewhat higher because (3) refers to tracks etched from the
middle towards both ends. A track that etches from one end towards the other requires a longer
etch time, during which it acquires a greater thickness; equation (6) is also based on rather *ad*
*hoc* estimates but the conclusion stands. **Figure 6e-h** shows, based on the same reasoning and
with the same proviso, that most tracks etched for an effective duration of ~10 seconds are
etched to their ends, which lends support to the (20 + 10)-s etching protocol proposed by Tamer
and Ketcham (2023).
We cannot, at this stage, estimate the precise effects of each constraint on the confined track se-
lection, but we can make a distinction between hard constraints (1) and (2) and soft constraints
(3) and (4), which depend on the track length, but have less effect on the selection. The threshold
width (2) can shift over a fraction of a micrometre depending on the operator, but the maximum
width (1) is a function of the immersion time and the anisotropic apatite etch rate. In conse-
quence, the confined track sample selected for measurement will depend on the immersion time
and etch rate, and thus on the etchant and apatite compositions. This means that the anisotropic
apatite etch rate is the main cause of the different angular distributions of confined tracks in ap-
atites with different Dper/Dpar ratios (Barbarand et al., 2003; Ketcham, 2003; Ravenhurst et al.,
2003). However, not because of a target-projectile relationship (Galbraith et al., 1990; Galbraith,
2002; Ketcham, 2003) but because of the variation of the apatite etch rate with orientation
(Aslanian et al., 2021).
A series of recent studies throw light on the influence of the etching protocol, experimental con-
ditions (transmitted *vs.* reflected light) and selection criteria on measurements of step-etched
confined tracks in Durango apatite (Tamer et al., 2019; Tamer and Ketcham, 2020; 2023;
Ketcham and Tamer, 2021). The overall conclusion is that substantial differences of the measured
mean track length arise from the etching protocol and selection but less from the actual measure-
ments. Individual mean lengths 10-20 standard errors from the overall sample means (Ketcham
et al., 2015) are interpreted as due to differences in the evaluation of the roundedness of the
etched track ends. This contrasts with the excellent agreement of our current measurements of
the same samples with the annealing models based on the calibration data of Carlson et al. (1999).
We submit that if track selection is indeed for the most part controlled by constraints (1) and (2),
with modifications from (3) and (4), as our single-track data indicate, then it must be possible to
perform reproducible and meaningful confined track length measurements which are suitable





for modelling. An unbiased systematic search for horizontal confined tracks using transmitted
light is adequate for this purpose. Deliberate or inadvertent biasing, carelessness or inexperience
will affect the results, but these should be treated as statistical outliers, not as an indication that
track lengths are fluid.
**Figure 6i-l** shows a weak positive correlation between the *c*-axis projected lengths and effective
etch times of individual confined tracks. Precise rates of increase are difficult to estimate because
of the weak correlation and large scatter of $t_E$ and $l_P$. The best estimate is based on the high-angle
tracks because their etch times are calculated from the greatest widths and a single etch rate
(**Figure 1f**; equation 1b). The result (2.3 μm/min) is somewhat lower than the value for fossil
tracks, which drops from 3.4 μm/min at 5 s to 2.9 μm/min at 20 s (Aslanian et al., 2021). It is
closer to the rate for the non-projected lengths of fossil and induced tracks reported by Tamer
and Ketcham (2020; 2.6 μm/min).

### 3.4 Track etch rate

The calculated track etch rates span an order of magnitude (60-600 μm/min) in all samples and
the $v_T$-distributions are right-skewed in each case (**Figure 7a-d**). This can be a measurement ef-
fect because $v_T$ is obtained by from the apatite etch rate $v_R$ perpendicular to the track margins and
the subtended angle $\theta$ (equation 3a). Because $\theta$ averages 3-4° and is difficult to measure, signifi-
cant errors of 1° and more cannot be excluded. In that case, the harmonic mean $v_T$ are less biased
central estimates than the arithmetic means. These range from ~160 μm/min for the unannealed
sample to ~120 μm/min for the most annealed. There is however no connection between the ex-
tent of partial annealing and the track etch rate. Given the uncertainties it is reasonable to sup-
pose that there is no demonstrable effect of annealing on the etch rate of induced tracks. The
overall mean (harmonic or arithmetic) for the four studied samples is ~140 μm/min. This is twice
the value for unannealed fossil tracks of Aslanian et al. (2021). Their arithmetic mean is 75
μm/min; the harmonic mean for the same data is ~63 μm/min. Even considering the uncertain-
ties, it appears that fossil tracks etch at a slower rate than comparable, partially annealed induced
tracks, which can be related to their different annealing behaviour (Wauschkuhn et al., 2015a).
Step-etch experiments also revealed differences between fossil and unannealed induced tracks
(Jonckheere et al., 2017).
Our calculation assumes that a single $v_T$ value characterizes the entire track (equation 3a; **Figure
1e**). This implies a constant-core $v_T$-model rather than a linear model (Tamer and Ketcham, 2020;
Ketcham and Tamer, 2021), one in which the core extends over most of the etchable track length,
except for perhaps ~1 μm at either end, where the average etch rate drops as the damage be-
comes intermittent (Paul and Fitzgerald, 1992; Paul, 1993; Toulemonde et al., 1994; Li et al.,
2011; 2012). This is at variance with the view that $v_T$ reflects the variation of the damage or di-
ameter along the track, resulting from its formation (Fleischer et al., 1969; Price et al., 1973). It is
nonetheless reasonable to consider that depleted, amorphous or porous track cores (Fleischer et
al., 1965; Szenes, 1996a; b; Li et al., 2010) could act as conduits along which the etchant advances
at a rate that does not depend on the degree of material depletion, disorder, or porousness of the
track core, but on the rate of fluid flow, the reaction rate, or the exchange rate of acid and reaction
products at the solid-fluid interface. This could explain the different (etchant-strength/etch-time)
protocols used for revealing fission tracks in apatite over much the same length, the revelation of
ion tracks from different particles with the same etchant, and the different shapes of etched tracks
produced with low and high acid concentrations (references in Jonckheere et al., 2017; Jonck-
heere and Van den haute, 1996).
Constant-core models avoid a number of problems of linear models, and those with variable $v_T$ in
general. Due to the "motorboat effect", linear $v_T$ models generate different-shaped confined tracks
depending on whether they are etched with the $v_T$-gradient, from their centre towards both ends,
or against the gradient, from one end towards the centre (Fleischer et al., 1969; 1975; Paretzke
et al., 1973). The track outlines in consequence exhibit concave and convex curvature not ob-
served in actual samples (**Figure 8**). The fact that confined tracks and surface tracks are straight

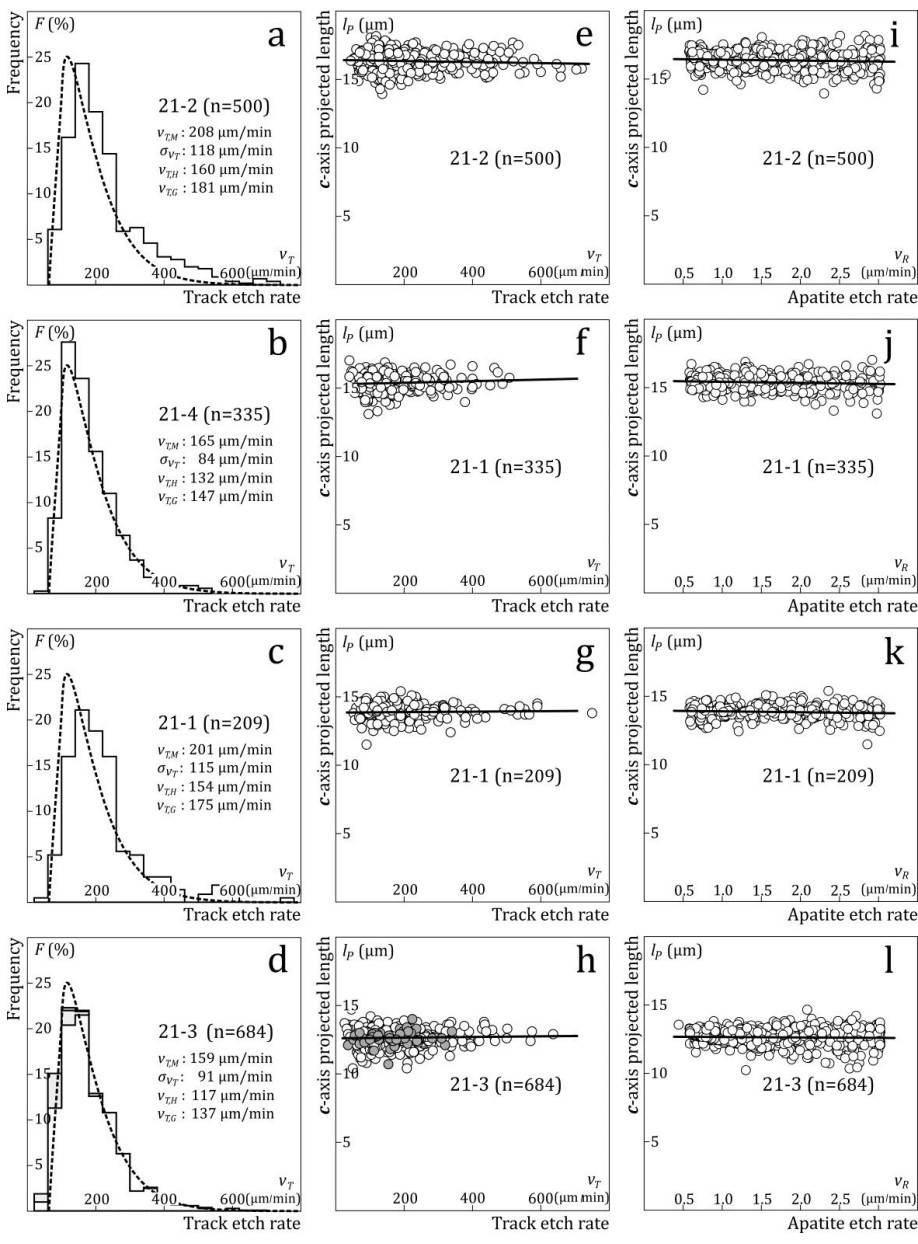

**Figure 7**. (**a-d**) Etch-rate ($v_T$-) distributions of horizontal confined tracks in the four studied samples; white: continuous tracks; grey: stepped tracks; black: gapped tracks; the dashed line is a fit to the combined data; $v_{T,M}$ and $s_{v_T}$: arithmetic means and standard deviations of the etch-rate distributions; $v_{T,H}$ and $v_{T,G}$: harmonic and geometric means; (**e-h**) $c$-axis projected lengths ($l_P$) plotted against the etch rates ($v_T$) of horizontal confined tracks ; white: low and moderate angle tracks (**Figure 1e**); grey: high-angle tracks (**Figure 1f**); the solid black lines are linear regression lines fitted to the combined data; (**i-l**) $c$-axis projected lengths ($l_P$) plotted against the apatite etch rates ($v_R$) in the direction of the tracks; the solid black lines are linear regression lines fitted to the combined data.





-edged irrespective of whether etching started from the centre or from the end is in itself a strong
indication of a constant $v_T$ (**Figure 9**). A linear $v_T$ model creates an excess of underetched tracks,
e.g., whenever etching starts at the end of a track or its effective etch time is less than the immer-
sion time (**Figure 8**). It is possible that underetched confined tracks are not observed or not se-
lected for measurement (Ketcham and Tamer, 2021), but this is not the case for surface tracks,
which are not selected. Nevertheless, all measurement of surface tracks reveal a deficit of short
tracks rather than an excess (Dakowski, 1978; Grivet et al., 1993; Jonckheere et al., 1993; Jonck-
heere and Van den haute, 2002). Surface tracks with lengths approaching those of confined tracks
also could not exist (**Figure 8d**). On reflection, there are other difficulties with linear $v_T$ models.
They predict that a substantial fraction of the confined tracks exhibit a rapid length increase at
the end of a standard immersion time, which is not observed in step-etch experiments (Jonckheere
et al., 2017; Tamer et al., 2019; Tamer and Ketcham, 2020, 2023). A linear model also implies that
a latent track cannot be etched over its entire length at the track etch rate $v_T$ in less than infinite
time as $v_T \rightarrow 0$ at its ends.
Our findings support a constant-core model extending over most of the etchable length of a track.
Nevertheless, $v_T$ must decrease towards the ends (Figure 8 of Laslett et al., 1984; Figure 1 of Watt
and Durrani, 1985; Figures 7c and 8c of Aslanian et al., 2021). In a constant core model, it is not
obvious that this decrease correlates with a gradual change of the latent-track properties. Latent-
track and etched-track studies (Paul and Fitzgerald, 1992; Paul, 1993; Li et al., 2011; 2012;
Wauschkuhn et al., 2015b; Jonckheere et al., 2017) indicate an intermittent structure, made up of
damaged sections etching at a rate comparable to $v_T$, alternating with recrystallized sections etch-
ing at a rate comparable to the apatite etch rate $v_R$ (unetchable gaps; Green et al., 1986). In that
case, the rate of length increase $v_L$ is an average of $v_T$ and $v_R$ over a short section of track. A de-
creasing $v_L$ towards the track tip causes it to become rounded (motorboat effect; Fleischer et al.,
1969; Ketcham and Tamer, 2021), until a gap cannot be breached and the end of the etched track
becomes bounded by basal and prism faces (Aslanian et al., 2021). The transition of rounded to
polygonal track tips would be the point at which an individual track is fully etched (Tamer and
Ketcham, 2023). Our results, however, reveal no correlation between the ***c***-axis projected lengths
and the track (**Figure 7e-h**) or the apatite etch rate (**Figure 7i-l**). This can mean that no correla-
tion exists or that it is weak and obscured by the statistical variation of the length and etch-rate
measurements.
Considering that a constant-core model implies that $v_T$ is insensitive to the properties of the latent
track, it is surprising that substantial differences appear to exist between individual tracks. At
this stage, we cannot exclude that most of the $v_T$-variation is related to its measurement and cal-
culation, and that induced fission tracks in apatite in fact have the same or a narrow range of $v_T$-
values. Even in that case, the harmonic means of our measurements provide valid first-order $v_T$-
estimates indicating that the cores of fossil and induced tracks etch at different rates. Price et al.
(1973) ascribed the different etching behaviour of "old" and "new" tracks in meteoritic minerals
to a rearrangement of the damage. In apatite, this phenomenon could be related to ageing (Gleadow
and Duddy, 1981) or seasoning (Durrani and Bull, 1987; Wauschkuhn et al., 2015a). However, the
significance of the core etch rates for the lengths of etched tracks is at present unclear (Ketcham
and Tamer, 2021).

### 4. Conclusions

Our investigation is concerned with the factors affecting the selection of confined tracks for meas-
urement and T,t-modelling. It was conducted on induced tracks in four prism sections of Durango
apatite etched for 20 s in 5.5 M $HNO_3$ at 21 °C. Their nominal mean track lengths are ~16, ~14,
~12, and ~10 μm (Ketcham et al., 2015), and their transmitted-light track densities are between
~2.9 and ~1.9 $10^6$ cm$^{-2}$ (Aslanian et al., 2022). The tracks were selected by scanning the samples
in transmitted light using two conscious criteria: (1) the entire track could be captured in an im-
age stack with a depth of 1.25 μm; (2) both ends of the track were distinct. One operator collected
the images; the second vetted them, without deliberating with the first, and performed the meas-
urements. In order to characterize each selected track as fully as possible, these included length,
orientation, width, cone angle and other dimensions needed for calculating their individual effec-
tive etch times.

**Figure 8**. Geometries of unannealed induced confined tracks in apatite etched 20 s in 5.5 M HNO₃
at 21 °C, calculated with the linear $v_T$-model of Ketcham and Tamer (2021; Figure 15 and Table 2)
(**a**) track etched from the middle towards both ends; (**b**) confined track intersected at ¼ length
from one end; (**c**) at ⅛ length; (**d**) at the end; brighter grey shades indicate increasing etch times
from 5 to 20 s.



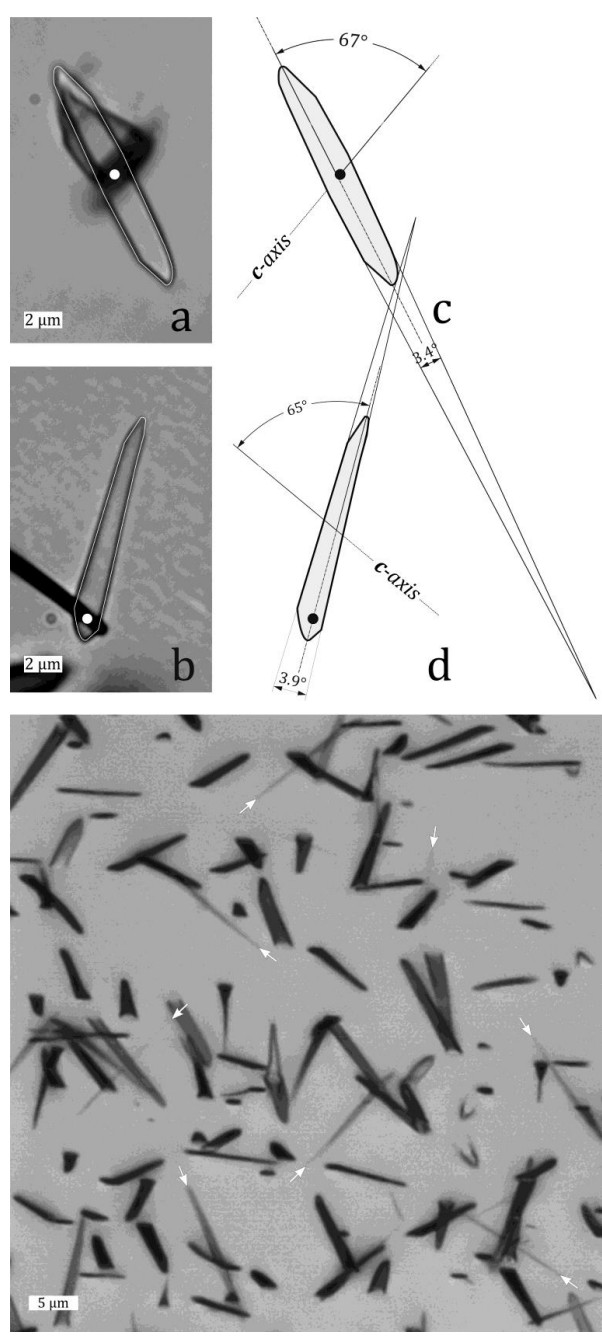

**Figure 9**. Microscope images and contours of confined tracks with a host track intersections close to the middle (**a, c**) and close to the end (**b, d**), both showing straight edges indicative of a constant $v_T$; (**e**) compressed image stack of straight-edged surface tracks in a prism face of Durango apatite etched 20 s in 5.5 M $HNO_3$ at 21 °C, showing no excess of short tracks or deficit of long tracks; the fact that all confined tracks, however thin compared to surface tracks with the same orientations, are etched over something approaching their full lengths also argues for a constant high track etch rate (arrows).





In contrast to the large variation reported in Ketcham et al. (2015), our results are in good agree-
ment with the annealing equations and original data for this apatite and etchant (Carlson et al.,
1999; Ketcham et al., 1999). This leads us to question whether all variation between analysts can
be reduced to individual judgements as to which tracks are suitable for measurement (Ketcham
and Tamer, 2021). Our data show that $c$-axis projection (Donelick et al., 1999) eliminates all track
length variation with orientation but leads to a narrowing of the length distribution with increas-
ing angle to the $c$-axis, with consequences for T,t-modelling that merit thorough consideration.
Our sample annealed to a mean length of ~10 μm contains segmented tracks, whose number in-
creases and whose length decreases with their angle to the $c$-axis, and thus with a decrease of the
apatite etch rate. Their appearance is due to a local obstruction delaying the progress of the etch-
ant, i.e., to unetchable gaps (Green et al., 1986) rather than accelerated length reduction (Donelick,
1991). Our measurements show that the gaps are between ~10 and ~100 nm, with a mode at
~30 nm, confirming estimates based on transmission electron microscope observations (Paul
and Fitzgerald, 1992; Paul, 1993).
The confined track selection is in the first place determined by a threshold width and in the sec-
ond place by the requirement that the tracks are etched to their ends. In most cases the first con-
dition implies the second, which decreases in importance are the tracks are shortened by anneal-
ing. The remaining cases correspond to orientations in which the track width increases fastest
compared to its length. The selection is further limited by the fact that the widest confined tracks,
which are also the shallowest, come to intersect the surface, which eliminates them from consid-
eration. This is a surface proximity bias with the emphasis on the width rather than on the length
of the tracks (Galbraith et al., 1990; Galbraith, 2005). The number of suitable confined tracks at
each angle to the $c$-axis is proportional to the difference between the minimum and maximum
widths corresponding to these constraints. The angular distribution of the selected tracks is there-
fore a close reflection of the range of possible track widths, with little influence from geometric
biases. Because the confined track selection depends on the track widths rather than lengths, it
is almost unaffected by annealing, at least up to the point that selective track loss occurs. A change
of the etch rates, due to the etchant or apatite composition would in contrast have a definite effect
(Ketcham, 2003).
We calculated the true duration for which each individual confined track has been etched (effec-
tive etch time) from its width and the apatite etch rate perpendicular to its axis (Aslanian et al.,
2021). The results remain consistent with the constraints, converted to etch times in the same
manner. These are somewhat indistinct because they depend to an extent on the length of each
confined track and on where along its length the host track intersects it. In favourable orienta-
tions, an unannealed track intersected in the middle can etch in five seconds. This places a lower
limit 100 μm/min on the track etch rate. The etch time distributions are right-skewed with a
mode at just over half the immersion time. The thinnest tracks at low and high angles to the $c$-
axis need the longest etch times and also appear to be less affected by surface proximity bias than
the wider tracks. The etch time distribution shows no demonstrable dependence on the extent of
annealing despite the different geometrical relationship between the unetched host tracks and
confined tracks. This illustrates a principle known from other selection processes, nl. that those
entities are selected that have the right qualities for being selected, in this case the track width is
the overriding condition.
Our calculated track etch rates span an order of magnitude, with arithmetic means of 160 to 200
μm/min and harmonic means of 120 to 160 μm/min. The $v_T$-distributions are right-skewed with
values ranging up to 600 μm/min, which are thought to be overestimates because the $v_T$-calcula-
tion is vulnerable to measurement error. In contrast to earlier reports (Tamer and Ketcham,
2020; Ketcham and Tamer, 2021), we find no evidence for an increase of $v_T$ with annealing or for
variation of $v_T$ along the tracks. We favour a constant etch rate over most of the track length be-
cause it seems inevitable that a linear model creates a great excess of short tracks, which even if
excluded from the confined track sample (Ketcham and Tamer, 2021), would produce a concave-
upwards distribution of the projected lengths of surface tracks (Laslett et al., 1982; Rebetez et al.,



1990), which is not supported by measurements (Dakowski, 1978; Grivet et al., 1993; Jonckheere et al., 1993; Jonckheere and Van den haute, 2002). Calculation also shows that for a linear $v_T$-model the etchant will often fail to reach the ends of the confined tracks within the standard immersion time (**Figure 10**). A constant-core model does not have this drawback; for the mean etch rates determined in this work (140 µm/min), it takes under seven seconds to etch the longest track from end to end. A high $v_T$ would also account for the common experience that almost all the confined tracks in a sample have close to their full lengths, and for the fact that the measured lengths exhibit no correlation with $v_T$. A high $v_T$ would also limit the effect of several host tracks intersecting a confined track, which we estimate to be negligible unless it is intersected at distant points at the same time. This implies that the dependence of the track length on effective etch time is a limited effect due to etching at its endpoints at a rate $v_L$ intermediate between the track and the apatite etch rates (Aslanian et al., 2021). The question of the track etch rate nevertheless presents challenges as it seems reasonable that if $v_T$ exhibits no variation along a track, it would also show little variation from track to track. Our measured $v_T$ also do not correlate with the extent of annealing of the induced tracks, but are nevertheless twice as high as the value for fossil tracks (75 µm/min; Aslanian et al., 2021).

**Supplement link**

**Author contribution**

RJ designed the experiment, collected the microscope images, analysed the data and prepared the manuscript. Dr. M.T. Tamer made a substantial contribution to the measurements but desires not to be listed as co-author.

**Competing interests**

The author declares that he has no conflict of interest.

**Acknowledgement**

Funded by the German Science Foundation (DFG grant JO 358/4). I am pleased to acknowledge contributions of Dr. M.T. Tamer, Dai Mengyao, Zhong Chang, Yaling Tao to the image processing and measurements.



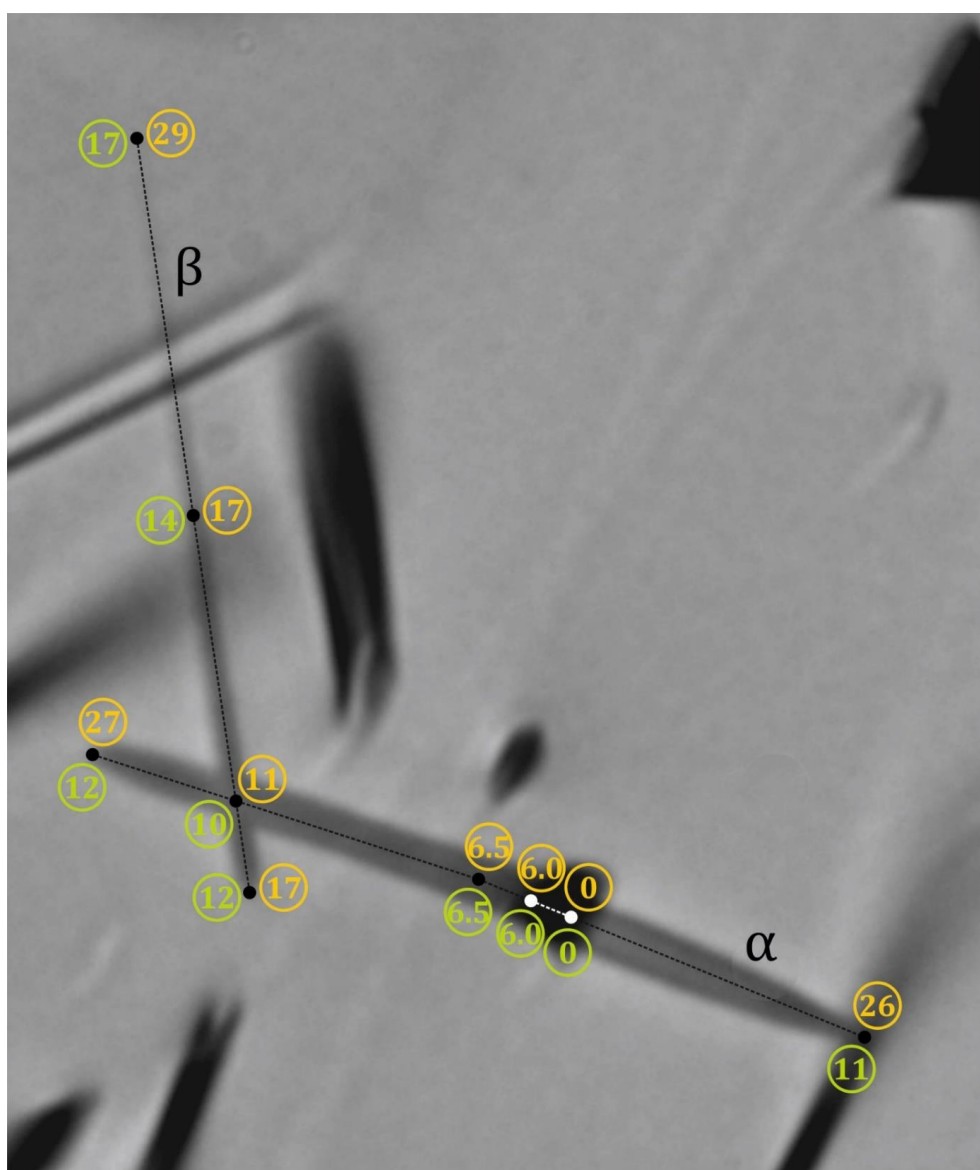

**Figure 10**. Comparison of etchant progress along an unannealed induced track-in-track-in-track in Durango apatite etched 20 s in 5.5 M HNO₃ at 21 °C. Yellow: seconds after immersion based on the linear model of Ketcham and Tamer (2021); green: seconds after immersion based on the effective etch times and track etch rates calculated from the measured widths using the etch rates of Aslanian et al. (2021). The etchant takes ~6 seconds to advance down the host track and across to intersect α; both models are in rough agreement about the time taken to progress from there to the intersection of α and β, and somewhat less along the main section of β. However, the calculations diverge towards the ends of both tracks due to the rapid decrease of the track etch rate in the linear model. Because of the expenditure of ~6 seconds of etch time before the etchant can begin to etch α, it is unable to reach the ends of α or β within the immersion time of the sample. According to the calculation based on Aslanian et al. (2021) the etchant reaches the furthest end of β with three seconds to spare, which is enough to widen it to ~0.2 μm and for it to be observed and measured (Appendix A).

# Appendix A: etch time calculation

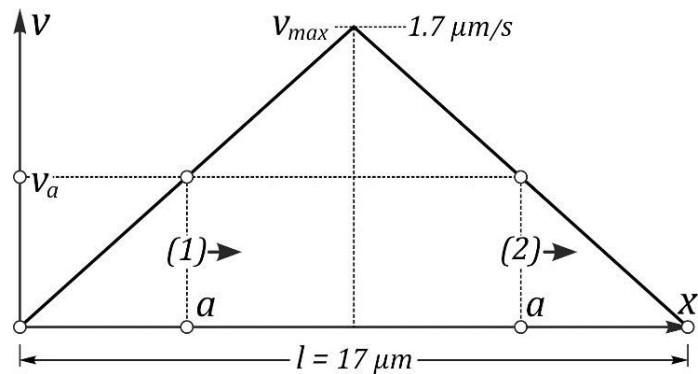

632
633
634

**Figure A1.** Linear etch rate model for unannealed induced tracks
(after Ketcham and Tamer, 2021; Figure 15); (1) etching against the
etch-rate gradient, (2) etching with the gradient.

$$\frac{dv}{dx} = c \qquad \text{(A1)}$$

$$\frac{dv}{dt}\frac{dt}{dx} = c \qquad \text{(A2)}$$

$$\frac{dv}{v} = c\,dt \qquad \text{(A3)}$$

$$\ln(v) = c\,t + u \qquad \text{(A4)}$$

$$t = 0 \rightarrow x = a \rightarrow v = v_a$$

$$\ln(v_a) = u \qquad \text{(A5)}$$

$$\ln\left(\frac{v}{v_a}\right) = c\,t \qquad \text{(A6)}$$

$$v = v_a\,e^{ct} \qquad \text{(A7)}$$

$$\frac{dx}{dt} = v_a\,e^{ct} \qquad \text{(A8)}$$

$$dx = v_a\,e^{ct}\,dt \qquad \text{(A9)}$$

$$x = \frac{v_a}{c}\,e^{ct} + z \qquad \text{(A10)}$$

$$t = 0 \rightarrow x = a$$

$$a - \frac{v_a}{c} = z \qquad \text{(A11)}$$

$$x = \frac{v_a}{c}\,e^{ct} + \left(a - \frac{v_a}{c}\right) \qquad \text{(A12)}$$

$$x = a + \frac{v_a}{c}\,(e^{ct} - 1) \qquad \text{(A13)}$$

---

$$(\mathbf{1})\ \ v_a = c\,a$$

$$x = a + \frac{c\,a}{c}\,(e^{ct} - 1) \qquad \text{(A14)}$$

$$x = a\,e^{ct} \qquad \text{(A15)}$$

$$t = \frac{1}{c}\,ln\left(\frac{x}{a}\right) \qquad \text{(A16)}$$

---

$$(\mathbf{2})\ \ v_a = c\left(\frac{l}{2} - a\right)$$

$$x = a + \left(\frac{l}{2} - a\right)(e^{ct} - 1) \qquad \text{(A17)}$$

$$t = \frac{1}{c}\,ln\left(1 + \frac{2(x-a)}{(l-2a)}\right) \qquad \text{(A18)}$$

---




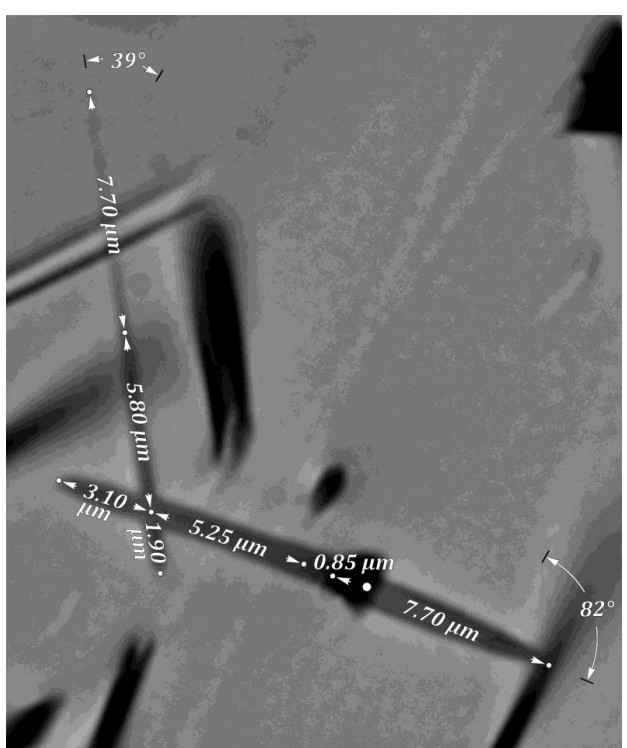

**Figure A2.** A track-in-track-in-track in an apatite prism face etched 20 s in 5.5 M HNO$_3$ at 21 °C, with the orientations and lengths of different sections indicated.

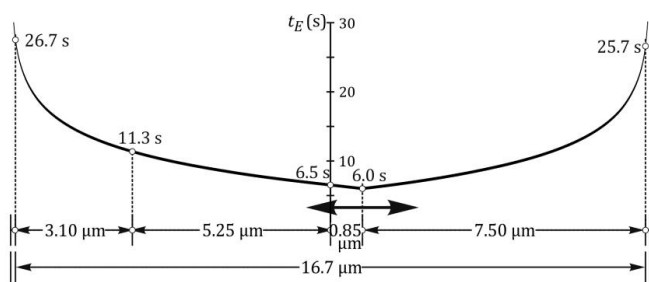

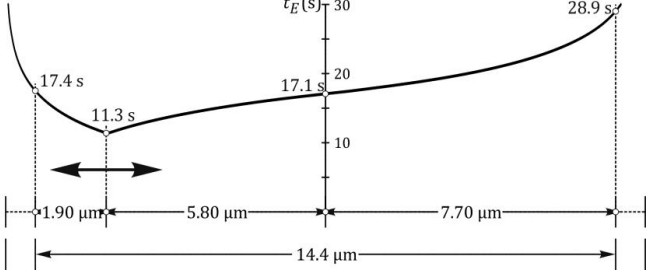

**Figure A3.** Calculated etch-time profiles of the tracks in Figure A2.

# Appendix B: *c*-axis projections

This appendix shows a comparison of alternative **c**-axis projections of confined track lengths.
**Figure B1** shows the results of the known method (Donelick et al., 1999), which is implemented
in programs for modelling thermal histories. It assumes that each measured track length
represents the mean of a different population. The projection in **Figure B2** assumes that each
measured length represents a different track from the same population distributed about a single
ellipse.

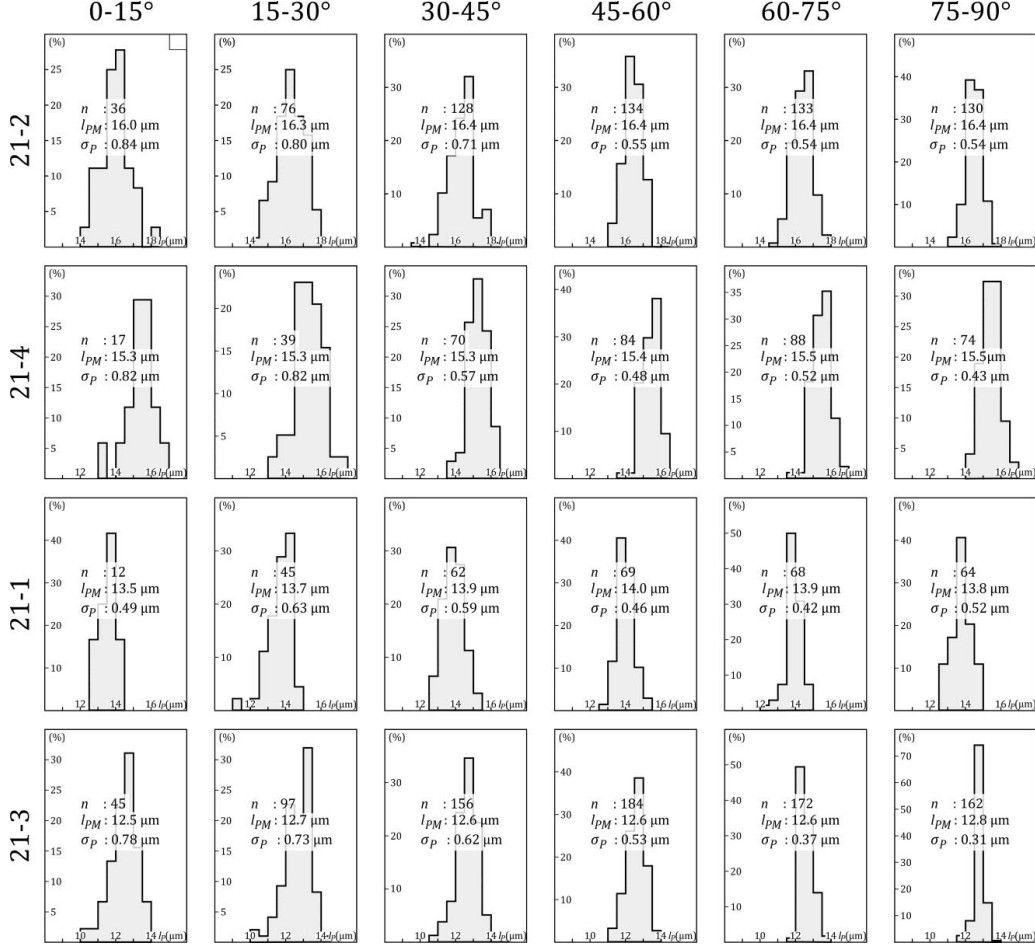

**Figure B1**. Normalized frequency distributions of the **c**-axis-projected track lengths in the four
studied samples for populations within different angular intervals (15°) to the apatite **c**-axis. The
different populations have consistent mean lengths but the standard deviations of the distributions show a marked decrease with increasing angle to the **c**-axis. This trend is clearest for samples 21-2 and 21-3, which are based on a greater number of measurements; $l_{PM}$: mean; $s_{PM}$: standard deviation.



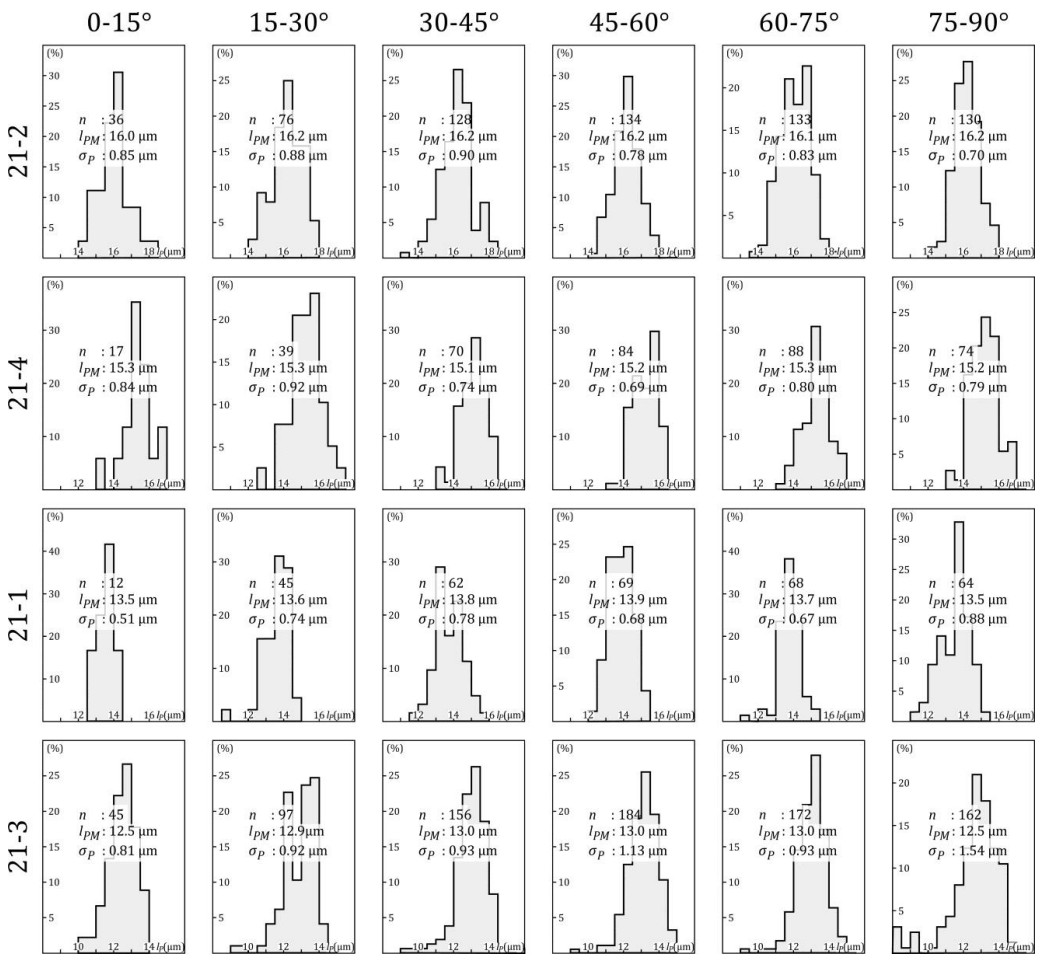

**Figure B2.** Normalized frequency distributions of the ***c*-axis-projected track lengths in the four
studied samples for populations within different angular intervals (15°) to the apatite ***c*-axis. For
comparison with **Figure B1**, the projection preserves the absolute differences between individual
lengths.





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
