# Peer review of "On etching, selection and measurement of confined fission tracks in apatite"

_Geochronology, 2023_

## Community Comment (CC2)

Geochron 2023-13

Response to R. Jonckheere (AC1)

Richard Ketcham and Murat Tamer

As with Dr. Jonckheere's judgement on our previous contribution to this journal (KT21), we recommend publication of this work, even if we have quarrels with it, though we hope more can be fixed as described below. Our preferred solution would be for a few assertions to be moderated, and for the applicability of MT's related measurements to be considered and discussed rather than outwardly ignored and implicitly contradicted, and the original authors come to mutual agreement about the content of the paper, rather than one insisting on his own way and brooking no dissent. Failing this (or either way, really), we would request that publication be conditioned on making all data available.

Following are some final responses for this discussion. Comments by the author are in red italics (except those in gold, where he was quoting our previous comments), and do not cover every point of discussion, but are presented in the original order.

*"I rederived the equations because I was interested in the preferred linear vT-model, and wished not to complicate the equations with vB, as vB > vT over 0.11 μm at the ends of unannealed induced tracks. I did, however, consider vB in my calculations by setting vT = max(vT, vB)."*

This accommodation is not sufficient to match the KT21 model, as it misrepresents the modeled etch rate along the entire track; the final 0.11 μm should not etch at vB, but at a higher rate. When graphed (right) the difference may seem insignificant, but in practice it makes a substantial difference in calculating length as a function of time. To use Dr. Jonckheere's shortcut, the latent length could be increased to make sure the etch rate matches vB at the same distance where the KT21 model does, in this particular case 17.22 μm rather than 17.0. In his response, RJ changes the c-value

(rate slope), but it's not clear if this completely corrects the problem; Figure 4 of the response remains incorrect, because the red curve should be at vB after 8.5 μm, not at some lower velocity that needs to be corrected.

*"For discussion, it is perhaps useful to explain the disagreement with TK20 and KT21. (1) The manner of our vT-measurements assumes that vT is constant along most of the length of a fission track. If it isn't, and vT varies along a track, as the linear model requires, then our data are meaningless."*

We agree that constant vT is an assumption, and we disagree as to whether it is a necessary or well-advised one. All the same, it does not render the data "meaningless", but a kind of average, depending on the points selected for measurement.

*"(2) In contrast to TK20 and KT21, our samples provide no evidence for an increase of the track etch rate vT following partial annealing."*

Whether the samples provide no evidence is not certain – the author's analysis presented may imply this, but the analysis is not exhaustive, and is affected by the preconceptions that have gone into it.  This is why we support publication, but with the requested condition that all underlying data be made available.

We note that MT also measured several hundreds of spontaneous tracks in Durango apatite as a part of this study, but for unknown reasons these data have not been included in this manuscript.  Ideally, these would be made available as well, or at least to MT for his work.

*"Our microscope resolution is ca. 0.2 µm but this is not the precision of our measurements. Resolution is the least separation between two dots that can be distinguished under the microscope. Figure 1 shows the blurred contour of a horizontal confined track, but circles drawn with care for measuring vT do not have precision errors of twice 0.2 µm on their diameters. It would appear that the error on a width is about the same as that on a length measurement."*

It is unclear what "can be distinguished under a microscope" means in this context; exactly what is being distinguished? We also feel that the lines drawn on top of the images does more to obscure than illuminate – they present an interpretation, but mask the actual image data.  We agree that the uncertainty in the measurement is about the same as a length measurement, but as a proportion of the quantity being measured it can be substantial.  For example, Carlson et al (1999) report measurement precisions of individual track lengths of ±0.15 µm, after a thorough self-examination using remeasurement.  This uncertainty is negligible for a track length of 16 or 8 µm, but much more substantial for measuring track widths on the order of 1-2 µm, and even more so for utilizing the difference between widths to calculate an angle.

*"Our data result from direct vT-measurements, compared to model estimates based on mean track lengths."*

Well, to be precise: these are not direct vT-measurements, but measurements of widths and distances that are used to calculate vT with a derivation that assumes it is constant, and interpreted with an elegant model of etching anisotropy that nevertheless may also embed assumptions of constant etching velocity.

*"The 20-30 s data of all the samples relate to the final ~1 µm length increase of the tracks. I[t] thus appears that most of the vT-model rests on the 10 and 15 s data in block ③."*

This interpretation of appearance is not accurate.  The 20-25 s and 25-30 s etch steps are also influential in that the non-annealed samples do not reach the bulk etch rate indicated by the annealed samples until after 20 s or even 25 s, depending on when measurement began.

*"The large difference between the 10s mean lengths of the annealed and unannealed samples underlies the assumed increase of vT after annealing, and a good part of the linear model. I added published and unpublished data for unannealed fossil and induced tracks etched for similar duration (Figure 2). The shortest 10 s length (12.5 µm), by Murat Tamer, is >2.5 µm longer than the values in TK20 and KT21. The*

*rest is much longer still, up to >16 µm after 15 s immersion.  This suggests that the 10 s data for the unannealed samples of TK20 and KT21 are unsafe."*

We are particularly troubled by the tenor of this comment, and similar ones in this response (e.g., "(wrong?)" in Fig. 4 caption), which is the main thing that necessitated this response.  As pointed out by KT21, early-step length measurements in step-etch experiments are inevitably on under-etched tracks, and so it's unclear what drives the choice as to which tracks to measure and which to bypass.  It is likely that for most analysts that have made such measurements, the criterion is simply the somewhat amorphous "best tracks that can be found"; this was the case for the measurements labeled MT13.  However, the goal of the TK20 study was to find as many tracks as possible so their individual growth trajectories could be traced.  This resulted in a much more thorough and meticulous search procedure, finding tracks at the edge of visibility only discernable by racking the microscope focus, or flipping through stacks of captured images (see video in TK20 supplemental material).  It is likely that no previous studies used such a protocol. Insofar as these barely-visible tracks were shorter, they lowered the mean and increased the standard deviation compared to what other analysts have previously reported.

We also note that MT used the same preparation procedures, and attempted to maintain the same selection criteria, throughout data acquisition for TK20, although he did vary his imaging conditions depending on whether he was measuring in the evening (red/black for induced tracks) versus daytime (white/black for spontaneous tracks).  Additionally, prior to commencing these measurements there was no expectation as to what they would reveal that could lead to implicit biasing.  We thus maintain that these are quality data, and any accusation that they are "unsafe" would need far stronger evidentiary support.

It is this undertone of distrusting MT's related and pertinent data that make the paper, as written, unsuitable for MT to participate as an author.  We still believe that this paper can be an excellent scientific contribution with MT getting the credit he deserves for performing all of the measurements, by removing these undertones and allowing for more flexibility, modesty, and completeness in how the data are interpreted and compared.

We further note that the points flagged by RJ are for experiments where some degree of constant-core etching structure was supported by KT21.  Assuming RJ approves more highly of the data for the annealed-apatite experiments (in block ①), those are the ones that were just as well or better fit by the linear model as a constant-core one.  However, even in this case we note that KT21 reports that a constant-core structure is permitted by the data (their Fig. 13); it's just that the data supported the simpler model, and that's what we reported.

Finally, the JB03 data plotted on Figure 2 of the response are for apatite from the Fish Canyon Tuff, which is more soluble than Durango apatite and thus not suitable for comparison in this context.  Some of the other points we cannot comment on, as we're not sure where they are from (RJ18, LS20, the second circle with the same color as LS20 at 15s with a shorter mean).

*"I submit that this is related to the substandard microscope images (Figure 2; Figures 9 of TK20 and KT21)."*

We have always used the original images without changing the colors in our papers. Conversely, Figure 2 in RJ's reply is a modified version of Figure 9 in TK20, where the track tips are even less visible than the original image. The intention of the original figure was to show the evolution of track etching with progressing etch times, and did not use this level of magnification for the measurements. The original figure of our paper, and the animation we provided, better represent our measurement conditions.

Nevertheless, it is indeed possible that better microscope images (mount preparation, polishing, etc.) would have revealed somewhat longer detectable lengths for some 10s tracks measured for TK20, which could in turn have affected how the KT21 model fit the data. However, we don't believe that this difference would have been enough to overcome the general conclusion that track etch velocities diminish toward their tips, and in particular that unannealed spontaneous and induced tracks tend to be under-etched after 20s, as that conclusion is based on the later etching steps. We also don't believe it would have affected the observation that unannealed track populations featured a higher preponderance of short early-etch tracks than annealed ones, leading to a lower mean length and slower implied etching, given our efforts to keep our data comparable among each other.

*"I also believe that the length distributions would have suited the intended purpose better than the mean lengths."*

This objection is unclear; KT21 did calculate length distributions, but used the means of the calculated distributions to summarize results and optimize model parameters. Length distributions are provided in the supplemental material.

*[line 18] It's unclear whether the measurements described here actually tested for variation along the track length; only **one rate** was measured per half-track in most cases.*
*I can be clear that no such test was performed. Because the tracks are straight, ten measurements will give ten times the same result.*

This statement inappropriately presupposes that the answer is known; it is circular reasoning. It is also absurd in itself, as the author acknowledges that the measurement uncertainty is substantial.

*[line 366] It seems like Tamer's step etch data can be used to test some of the implied assertions of the **model** in Fig. 6a-d. Is it really tenable that tracks at ~70 degrees are not measurable at effective times >13s when you can measure them after a 10s step, with some as long as 13 um (after only 3s of etching)? The model described here predicts that there should be a large deficit of tracks at 60-75° if searching for them at 10s and verifying that they are still present after 20s, but that is not the observation in Tamer and Ketcham (2020). Also, again, are tracks with effective etch times of over 18s realistic given the need to penetrate the polished surface (how deep are these >18s tracks)? This is another case where it could be interesting to check track depth varies with c-axis angle.*
*Let me repeat that our length data are within ca. 0.1 μm of their predicted values. Unless one has concrete reasons to question our measurements, **the data are the facts**. That is how things are; whether that fits one's preconceptions is something each has to examine for himself; however, I would not question the data first.*

We do not disagree, but reiterate that TK20 report data, data measured by the same person who made all of the measurements in this work. Those are facts as well.

To clarify on the scientific point, if one can measure a 70° track after 10s, and then measure the same track after 20s, 25s, and 30s (as TK20 did), then it seems highly likely that its effective etching time exceeds the ~13s allowed by RJ's model (or, "tentative interpretation of actual data") in Fig. 6a-d.

*"Forgive me if I am growing impatient with the insistence that our results should be measured against the TK20 and KT21 "benchmarks"."*

We don't know where this statement comes from. We never use the work "benchmark," and never imply that these results should be "measured against" prior ones. However, all scientific reasoning is always improved by considering applicable, independent work that can corroborate or contradict. Moreover, the "model" here is one that the author is already mentioned many times, so it seems fair game to propose an actual to-the-point test for it, rather than relying on assertions.

*"Is it not established procedure to evaluate a model against the data instead of vetting the data based on a model? When there is disagreement, should we uphold the model and reject inconvenient data? Really?"*

Again, the proposal is not to vet the data against the model, but to look at the data in available, more pointed ways to see whether it contradicts (and heck, possibly disproves) the model. Concerning rejecting inconvenient data, we point again to the TK20 data (2 comments ago), which may or may not be inconvenient to RJ's thesis, but deserves mention, consideration and discussion in a paper where all measurements were made by the same individual.

*"I have a related question: in TK23 polygonal track tips appear shortly after ~10 s "effective etch time". How were the effective etch times determined and how does this square with a linear vT model and isotropic vB (TC19; TK20; KT21)? Is this not as flagrant a contradiction between model and observation as Figure 8?"*

We need this question to be much more specific to address it; does it concern the new data in TK23 (annealed tracks, with more complete experiments than in TK20 and giving an overall result consistent with his others), or the schematic diagram in its Fig. 1? If the latter, it is based on MT's own observation of thousands of confined tracks; the diagram is **schematic** in many ways, and does not take into account all applicable variables, such as mean etching rate and track length, where indeed one would expect shorter and/or faster tracks would reach polygonal tips in less time.

*[line 478-480] … for example, the white outlines in Fig 9a,b are sometimes on the inside edge of the blurred region, sometimes on the outside edge. One could draw a curved line on the left side of Fig 9b that is a **scaled version of 8c**.*
*No.*

Unnecessarily curt.  For the sake of dialogue, we attempt to roughly illustrate the point.  The first diagram to the right is Fig. 9b from the manuscript, with contrast adjusted to emphasize the blurred track edge and interpretive line.  If the track were truly straight, it should be possible to draw a straight line that shows the same amount of blurring to either side, or in other words to have the blurred track edge be symmetric about the straight line.  This appears hard to do; starting from the bottom and going up, the right-side line begins to the right middle of the blurred region, strays to the right edge, and ends up back toward the middle, implying a very slight curvature.  Similarly, the left line also cannot stay squarely in the middle of the blurry track boundary, appearing to go center-left-center-left.  On the right, we contrast this with a version of the outer modeled track in Fig. 8c, blurred to approximate the appearance of a track under a microscope.  Here, after blocking out the part intersected by the etchant channel, we can also draw a straight line along the simulated track that appears to stay in the middle, and only close inspection shows that it subtly varies its position within it.

[Figure]

Now, these possible appearances of non-straightness in the real track image may be a manifestation of other phenomena, such as refraction, 3D etched track shape, imperfections in the microscope image, variation in track depth, etc.  However, it seems clear that very little curvature is really implied by variable along-track etching rates, and within the context of the blurring and noise obtained from even high-quality optical microscopy, it could be missed, especially if one is convinced ahead of time that it is not there.

Lastly, one new question: are the "s" measurements (e.g., Fig. 1e) the distance between the centers of the circles, or along a line that is tangent to both circles?  We had thought the former, but then it seems that Equation 3b should use an arctan, not an arcsin.  Or, perhaps there is something else we are not understanding about the equation.

Richard Ketcham and Murat Tamer

---

## Author Comment (AC1)

**On etching, selection and measurement of confined fission tracks in apatite**

CC1: *comments. by Richard Ketcham, July 12 2023*

RC1: replies by Raymond Jonckheere, July 26, 2023

*This is a very interesting study that builds upon previous innovative work by the author and his student. Utilizing track shape is an extremely promising way to wrest more information out of fission-track data. As the technology of data acquisition and image analysis continues to progress, some shape data will undoubtedly become routinely available with little effort, and can certainly be utilized. That time is not quite here yet, but the author's group has invested considerable effort in try to build the foundations for what is to come, both observational (Aslanian et al. 2021) and theoretical (Jonckheere et al. 2022). The future of fission-track analysis will certainly include aspects of this work.*

I am obliged to Dr. Ketcham for his interest in our work and for his critical and constructive comments. It is useful to introduce some abbreviations: JT17: Jonckheere et al. (2017); TC19: Tamer et al. (2019); JA19: Jonckheere et al. (2019); TK20: Tamer and Ketcham (2020); KT21: Ketcham and Tamer (2021); AS20: Aslanian et al. (2020); AS21: Aslanian et al. (2021); JA: Jonckheere et al. (2022); TK23 Tamer and Ketcham (2023).

*That said, there are some critical errors in the present manuscript, which in my opinion make it unsuitable for publication at this time. First, the author attempts to compare his **results** to the (possibly competing) predictions of the variable along-track etching rate **model** of Ketcham and Tamer (2021), but rather than using the **equations** therein attempts to rederive them anew. In doing so, he made a **critical error** by taking the etch rate at the track tip to be zero, rather than the bulk etch rate. Simply put, his equations do not correctly reflect my model, rendering a number of plots and assertions incorrect.*

I rederived the equations because I was interested in the preferred linear $v_T$-model, and wished not to complicate the equations with $v_B$, as $v_B > v_T$ over 0.11 μm at the ends of unannealed induced tracks. I did, however, consider $v_B$ in my calculations by setting $v_T = \max(v_T, v_B)$. I included the derivation in appendix A to allow those who wished to check my results and for possible use in future calculations. I did not draw the "$v_B$-wings" in Figure A1 but neither did KT21 in their Figure 9. Figure 8, does have (exaggerated) $v_B$-wings and shows that I included $v_B$ in the calculation. The circular feature at the left of 8d is due to an isotropic $v_B$; the spike to the right grows first when it extends into higher $v_T$-values along the track.

*Second, he fails to "remove the log in one's own eye before removing the mote from his brother's" in not investigating the limitations of his own data. The wavelength of **visible light** (0.4-0.7 μm) induces an unavoidable limit to the precision of optical microscopy data, which is not too influential for track length measurements but is a much larger issue for track width, and larger still for measuring angles from which to infer etch rates. Thus far I'm not aware of any attempt to quantitatively investigate the **uncertainties** of these data, either by mathematical analysis or brute force repeated measurements. This makes it very difficult to critically think through the implications of these data. In addition, it appears that each **half-track** is measured once, and so it's unclear how the author would detect a change in along-track etch rate.*

For discussion, it is perhaps useful to explain the disagreement with TK20 and KT21. (1) The manner of our $v_T$-measurements assumes that $v_T$ *is* **constant** along most of the length of a fission track. If it isn't, and $v_T$ varies along a track, as the linear model requires, then our data are meaningless. (2) In contrast to TK20 and KT21, our samples provide no evidence for an **increase** of the track etch rate $v_T$ following partial annealing. I believe that scientific progress is better served by addressing disagreements than by ignoring them.

Our microscope resolution is ca. 0.2 μm but this is not the precision of our measurements. Resolution is the least separation between two dots that can be distinguished under the microscope. Figure 1 shows the blurred contour of a horizontal confined track, but circles drawn with care for measuring $v_T$ do not have precision errors of twice 0.2 μm on their diameters. It would appear that the error on a width is about the same as that on a length measurement. I believe our length data show that these are accurate.

That does indeed not prove that our $v_T$-measurements are precise, far from it, but we have thousands of tracks. Our data result from direct **$v_T$-measurements**, compared to **model estimates** based on mean track lengths.

We measured straight sections, often across the supposed "$v_T$-maximum", which are much longer than half a track (Figure 1).

**Figure 1.** A straight track can look crooked but it is difficult for a crooked one to appear straight.

[Figure]

*Third, the author neglects to discuss data that appear plainly **contradictory** to his assertions. Tamer and Ketcham (2020) present clear indications of changes of track etch rate, such as **annealed tracks** being longer than non-annealed ones after 10s of etching, as well as the overall sequence of lengthening over a step etch experiment. As discussed in the detailed comments below, the model proposed here makes a number of further predictions about minimum or maximum track etch times that appear well tested and definitively excluded by the Tamer and Ketcham (2020) results. Given that the same individual (M.T. Tamer) produced both data sets, the **scientific process** suggests that this evident disparity in results at least be discussed, and credible ideas offered on where the problem might be, or what alternatives are available. For example, it may be plausible that the huge variation in track etch rate inferred from these measurements could be from measuring different sections of tracks – arguably a more parsimonious explanation than asserting that etch rates vary from track to track by a factor of 10. Or, if Tamer's previous data are unreliable, where might that have stemmed from?*

I thought that not emphasizing that our data contradicted TK20 and KT21 would avoid discord. Several factors indeed convince me that their conclusions are not accurate. The fitted data (TK20 Table 2; KT21 Table 1) fall into three blocks (Figure 2). Block ① are experiments aimed at determining $v_B$; the 10 s lengths are relevant to $v_T$ but the longer etch times are not. Block ② shows 20-30 s data, commonly interpreted as bulk etching, but here recast as $v_T$, which is admissible. However, the modelled lengths ($L_{lat}$) of the three annealed samples are identical with their 20 s measured mean lengths. It follows that their 25 and 30 s lengths are due to $v_B$, not to $v_T$. The 20-30 s data of all the samples relate to the final ~1 μm length increase of the tracks. I thus appears that most of the $v_T$-model rests on the 10 and 15 s data in block ③.

The large difference between the 10s mean lengths of the annealed and unannealed samples underlies the assumed **increase of $v_T$** after annealing, and a good part of the linear model. I added published and unpublished data for unannealed fossil and induced tracks etched for similar duration (Figure 2). The shortest 10 s length (12.5 μm), by Murat Tamer, is >2.5 μm longer than the values in TK20 and KT21. The rest is much longer still, up to >16 μm after 15 s immersion. This suggests that the 10 s data for the **unannealed** samples of TK20 and KT21 are unsafe. I submit that this is related to the substandard microscope images (Figure 2; Figures 9 of TK20 and KT21). I cannot resist adding that this puts the preceding comment about the measurements in our manuscript in a somewhat different light. Carl Sagan said that "*extraordinary claims require extraordinary evidence*". In my opinion the evidence is insufficient. I

also believe that the *length distributions* would have suited the intended purpose better than the mean lengths.

A formal point: our data consist of >2000 direct **$v_T$-measurements** (track images can be submitted for inspection). These data contradict **model predictions** based on 27 mean track lengths. Does the scientific process not require that the model must account for the misfit with the data, rather than the other way around?

[Figure]

**Figure 2.** Left: images of unannealed induced tracks etched for 10 s (TK20); right: breakdown of the mean length data from TK20 to which the TK21 $v_T$-models are fitted. Additional data: MT13: Tamer (2013); RJ20: Jonckheere (2020); LS20: Sarkosh (2020); PG86: Green et al. (1986); CA: Aslanian (2020); JB03: Barbarand et al (2003); WC99: Carlson et al. (1999).

*The Ketcham and Tamer (2021) variable along-track etching model is very simplified because it is based on mean confined length data from step-etch experiments, and with few data points to fit one can only test a simple model. That said, the constant-core etch rate model preferred by the author was the first one I derived and tested because I thought going into it that it would be the correct answer. It was the exercise of actually trying to fit the Tamer and Ketcham (2020) data (replicated in Tamer and Ketcham 2023, Chem. Geol.) that pointed to the linear model as adequate for most (but not necessarily all) levels of annealing. Shape data promises to be an excellent independent or complementary data source for deciphering track structure amidst the uncertainty stemming from not knowing when any given track starts etching, and it may well allow development of a more detailed and comprehensive model closer to physical reality. The author may be repeating my mistake in believing he knows the answer ahead of time, and thus limiting his field of consideration.*

All models are simplified. KT21 (p. 438) "... *we neglect length and etching anisotropy* ...". Therefore all tracks have the exact same etchable (latent) length and all differences arise from when and where the confined tracks are intersected. This produces a continuum of track lengths between zero and the maximum length (I have never seen it). The greater part is culled using an operator bias function (evidence?) or tip roundedness (isotropic $v_B$!). That is fine, we gain valuable **qualitative** insight into various aspects of track etching from the most approximate modelling. But, I find it incomprehensible that one can expect to make accurate **quantitative** predictions, as claimed in these comments. I also do not see how TK23 supports the linear model since it shows polygonal track terminations (JA19; AS21) after ~10 s "effective etch time".

*Finally, I note that the acknowledgements state that M.T. Tamer "made a substantial contribution to the measurements but desires not to be listed as a co-author." I gather that this was because he was being asked to be on a manuscript that both contradicted and ignored his own measurements, putting him in an impossible place. This is a shame, and should be remedied if possible.*

I invited Murat Tamer to join me in an experiment which I had started (I had made the images), and taught him how to measure widths, cone angles, etc., all of which he did fast and well. After rather more work than I had reckoned, I submitted a manuscript with Murat as co-author. Against my advice, he requested the editor not to be listed as co-author. There was never a scientific disagreement, but I understand that there was a "conflict of interest", as described, i.e., co-authoring a paper contradicting an earlier paper with a different collaborator. Such events are unpleasant and, with age, I have become rather impatient, even intolerant, of research that is not in the time honoured tradition, but political and calculating. I would be glad if Murat Tamer would be co-author of this manuscript again if he has changed his mind, agrees with the content, and expresses the wish to do so. I understand that it requires no resubmission.

Detailed comments:

*[line 10] The statement that the widest tracks "must also be the shallowest" is incorrect in two ways. First, width depends heavily on crystallographic **orientation**, and so one needs to consider this within angular bins of some sort. Second, within a given angular bin, width depends on time of intersection by the etchant channel (i.e. effective etching time). A shallower track is more likely to be intersected early, but given a limited number tracks within a bin, one or more **deeper** tracks easily could be intersected earlier, and thus be wider. (Gleadow, 1980: but very important restrictions are imposed by the effects of anisotropic etching).*

No, it isn't; the shallowest tracks are not per se the widest, but the **widest** tracks are the **shallowest**. I had imagined that I had earned the privilege not to be lectured at on track width, anisotropy, or effective etch time.

*[line 18] It's unclear whether the measurements described here actually tested for variation along the track length; only **one rate** was measured per half-track in most cases.*

I can be clear that no such test was performed. Because the tracks are straight, ten measurements will give ten times the same result. What is the meaning of suggesting that we measure the variation of $v_T$, while implying that we cannot measure a one rate. What is the point of disputing evident track shapes to defend a mere model? I show straight tracks in the manuscript and in Figure 1, and can produce several thousand more images showing the same thing. What do these comments have to offer, apart from innuendo?

*[line 102-103] Indeed, this is a significant bias. It might be clearer to just compare this to a 16-micron track (4.4 degree dip). It probably only has a small effect on this study, but is not something one would want to do when measuring unknowns.*

Our data did not reveal a selection or measurement bias that might be related to the dip of the tracks.

*[Figure 1] Needs scale bar or statement of image width.*

Right.

*[line 120] How do you know if there was a gap if you did not pierce it? It might eb better to state this as an interpretation, rather than an observation.*

How do I know that the thing is a track at all? I see no reason to question the interpretations of Green et al. (1986) and Galbraith et al. (1990). In this case the absence of a normal polygonal termination is evidence enough.

*[line 130] Again, if only one pair of measurements is made for a half-track, it's not clear whether or how that permits a change in etch rate along the track to be detected. It's also not clear how reproducible these measurements are with respect to (a) where one places the two circles along the track, and (b) how precise the circle margins are given the limited resolution of these measurements imposed by the wavelength of light. For example, some circles in Figure 1 go out to the edges of the blurry (resolution-limited) track*

*boundaries, while others are set noticeably within those boundaries. How does this affect the rate measurement? Has a **multiple-measurement**, come-back-to-it-later study been done?*

None of that! Is it not more productive to get a grip on the basics than to indulge in virtuous statisticulating? Our manuscript stresses the fact that almost all our mean lengths and standard deviations are within <0.1 μm of their predicted values. In contrast, I understand KT21 (Figure 17) to mean that one can measure whatever one likes. Are we now debating how many circles fit on the edge of a track? Is this musical chairs?

*[line 154] It's unclear what is meant by one "participant" versus the other, and what they both did. Did two people make all of these measurements (thus providing a **repeatability** analysis it would eb good to report), or did one make all of the measurements and the other just check to make sure they looked OK?*

I made the images and Murat Tamer selected and measured the tracks he considered suitable for length measurement. Does it matter? Our manuscript explains that "... *the present is a one-way rate concerning a single set of images using one set of etching and observation conditions*.". His overall rejection rate was <1%. Have the track length measurements in TK20 and KT21 been repeated before a model was fitted to them?

*[line 204] Unclear what is meant by "both projections", and which is the former versus latter.*

The former projection is shown in Figure B1, the latter in Figure B2. I will add more explanation to avoid confusion.

*[line 242] Not really; one must impose additional assumptions concerning **monotonicity**, or let the uncertainty going back in time propagate to very large values.*

This refers to: "*This (assumption implicit in $c$-axis projections) allows to convert its (a sample) age and length distribution directly to a Tt-path, without the need to search Tt-space*". This describes something I implemented in a thermal history program. If all measured lengths are interpreted as mean lengths and mean lengths shorten but do not ever increase, the order of mean lengths is also the order of formation. Yes, the errors increase as the number of tracks that experienced earlier temperatures grows smaller further back in time.

*[line 245] The reference is an abstract; I guess one can use it to claim that someone once said uncertainties can be taken into account, but it's not a source of information for how to go about it. In any event, how can there be a **single solution** that is faithful to the uncertainty in both the length measurement (both from the measurement and from natural variation) and the time intervals (which are certainly not even)? This paragraph seems to drift off-topic.*

I agree about the reference, but I have nothing else. I will remove it, as the statement does not require much support.

One solution is no solution. However I found that, without exception, it was the backbone of a set of solutions consistent with the data within statistical limits. If it is off-topic, it is because I am astonished to realize that an assumption that I made for convenience 32 years ago is imbedded in modern modelling software. I thought it worth mentioning because at the time (like a forward model) it was the starting point for a random search or for a perturbation method in which either all or selected nodes were allowed to wander within given limits. Perhaps it has some value for alerting trackers to the consequences of $c$-axis projection, for thinking of other algorithms for dealing with anisotropic lengths, or for developing new software.

*[line 258-261] A strange statement; you've already corroborated that anisotropy is removed well at varying levels of annealing. The lengths measured for a given annealing experiment project to a narrow range, but the means are significantly different at different levels of annealing. The original length and orientations are what the projection is based on – they must matter, or else the distributions of high-angle tracks at each annealing level would not be so narrow.*

This is an observation that can also be made from Figure 7 of Donelick et al. (1999). In the extreme case tracks perpendicular to the *c*-axis, their lengths, e.g., in the interval 4.0-10.4 µm are funnelled into a 1µm *c*-axis length interval. Length differences and measurement imprecision are therefore compressed by a factor of >6 (Figure 3). This does not contradict that *c*-axis projection is effective at eliminating length anisotropy.

**Figure 3.** Illustration of the compression of the range of lengths of high-angle tracks due to *c*-axis projection (after Donelick et al. 1999).

[Figure]

*[line 270-271] An alternative possibility could be that the section designated $l_3$ may not always be completely etched, due to the delay.*

Indeed, our data show that that this is the case (Figure 2d). But the difference between the total length ($l_T$) and that of continuous tracks is so small that we may conclude that no substantial track section is missing (Figure 4a).

*[line 275] It would be worth comparing this result with the recent report by **Li et al. (2003, EPSL)** of a gap at the center of each track upon formation. It's worth pondering whether there is really only one gap, and it's always at site of the original fissioning nucleus.*

The constriction resulting from track formation can be real or transient, but it has no effect on etching, or we would have seen it. Our data show that when it comes to annealing, the locus of the fissioned atom is not a preferred site for forming gaps (Figure 4c). I even suspect that our data are biased against gaps near the ends.

*[line 317-324] Another apparent oddity is that some many tracks at 0-20° and 85-90° apparently need to etch for the full **20 seconds**, which should be rare to impossible.*

Tracks sub-parallel and sub-perpendicular to *c* widen at the slowest rate ($v_R$) and thus require the longest etch times ($t_E$) to get over the threshold width ($w = v_R \times t_E$). In consequence, their etch time windows are narrow, meaning there are fewer of them, thus explaining the minima in the angular frequency distributions. In fact the range of widths ($\Delta w(\phi)$) between the threshold and the maximum is proportional to the relative angular frequencies ($F(\phi)$; Figure 4e-h). We do not measure the tracks that didn't make it, but those that did.

*[line 321-324] It's not clear whether this explanation makes sense; because etching along the penetrating channel is fast, there should be plenty of time for **deeper tracks** to grow wide and not be affected by the surface. This might be tested with the author's data – is there a correlation between track depth and width, or track depth and angle? This explanation predicts that tracks at 60-75° should be deeper, on average.*

This comment refers to the lack of tracks above constraint (4) in Figures 5a-d. There can be no discussion that they are the widest, as the vertical axis shows their measured widths. The question is why are there so few? What does it take to be overall the widest track? (1) It has to have the highest rate of widening (orientation). (2) It has to have the longest effective etch time, i.e., the shortest access time (immersion

time - effective etch time). How can it have the shortest access time: how else than by being closest to the surface (assuming that the time for the etchant to bridge the gap between host track and confined track is not dependent on depth, or also increases with depth because the host track is widest at the surface)? One can also drop the first condition and conclude that the widest tracks in a given direction are closest to the surface.

We did not measure the depths; I understand that Murat Tamer has some data and further plans in that direction.

*[Figure 5, lines 327-328] The y-axis in these figures is **frequency**; it's not clear what they have to do with delta-w. Were the wrong plots put into this figure?*

The frequencies refer to (1) the histograms, (2) the long dashed line which is a fit to the combined frequencies for all samples, and (3) to the short dashed line which predicts the angular frequencies from the range (spacing Δw) between the most restrictive constraints in Figure 5a-d. Therefore, one axis fits all.

*[line 342-343] Do **longer tracks** attain a greater width before they reach their ends? Figure 5a-c seems to contradict this – tracks are shorter at all angles as annealing progresses, but average widths seem to increase.*

Yes they do. It is best to compare 5a and 5d which both have more than twice the number of tracks of 5b and 5c, and the greatest length difference. One should reflect that (3) is not a sharp boundary but depends on whether a confined track is intersected in the middle or at the end. At this stage I believe it is relevant that eq. (6) offers a first-order explanation for a phenomenon that was not even known before our width measurements. It is not helpful to obsess about details near the detection limit, unless one has a better explanation.

*[line 366] It seems like Tamer's step etch data can be used to test some of the implied assertions of the **model** in Fig. 6a-d. Is it really tenable that tracks at ~70 degrees are not measurable at effective times >13s when you can measure them after a 10s step, with some as long as 13 um (after only 3s of etching)? The model described here predicts that there should be a large deficit of tracks at 60-75° if searching for them at 10s and verifying that they are still present after 20s, but that is not the observation in Tamer and Ketcham (2020). Also, again, are tracks with effective etch times of over 18s realistic given the need to penetrate the polished surface (how deep are these >18s tracks)? This is another case where it could be interesting to check track depth varies with c-axis angle.*

Let me repeat that our length data are within ca. 0.1 μm of their predicted values. Unless one has concrete reasons to question our measurements, **the data are the facts**. That is how things are; whether that fits one's preconceptions is something each has to examine for himself; however, I would not question the data first.

Figure 6a-d shows that tracks at ~70° are measurable from 5 to 13 s, including at 10s. The proposed test is indeed interesting. However, we invested considerable effort in single-track step-etching, and I am loath to go back to mean lengths, or even length distributions, plotted against immersion times when we have effective etch times. I am the first to admit that our data are noisy, but that does not make them wrong. I believe that experience has shown that 10 s immersion times are worse than useless for Durango apatite (Figure 2).

I beg not to refer to our work as "assertions" based on a "model", which I find abhorrent; the constraints are tentative interpretations of actual data that make sense to me, and are here presented to the reader for discussion.

*[line 432] It seems worth asking whether track rates really vary by a full **order of magnitude**, or the uncertainty of the rate estimation has something to do with it, or possibly because track etch rate varies and they are trying to measure etch rates along different sections of tracks. For example, a prediction of the Ketcham and Tamer (2021) model is that the etch rate for the $s_1$ track sections should be faster than for the $s_2$'s; this seems to be the case in Fig. 1g at least. Plotting these against each other seems like an easy test to try.*

Probably not. As the manuscript explains (lines 433-437), a large part of the variation is likely due to the statistical error on the cone angle measurement, which appears in the denominator of the $v_T$-equation. Hence the overdispersion and the right-skewness of the distribution, and why we recommend the harmonic mean.

Forgive me if I am growing impatient with the insistence that our results should be measured against the TK20 and KT21 "benchmarks". The model for unannealed induced tracks is based on eight mean track lengths plotted against immersion times; one measurement is questionable and another has been excluded. The data and model predict (1) an isotropic apatite etch rate, (2) tracks that have no intrinsic lengths, (3) length distributions that are the exclusive outcome of etching, (4) accelerated etching after annealing, (5) a linear $v_T$-model dependent on an *ad hoc* observer bias and an invalid tip roundedness criterion, all on the merit of $\chi^2$-values that can barely distinguish between a constant-core and a linear model.

Is it not established procedure to **evaluate a model against the data** instead of vetting the data based on a model? When there is disagreement, should we uphold the model and reject inconvenient data? Really?

*[Figure 8] This figure demonstrates a **mathematical error** in how the author has tried to reproduce the model of Ketcham and Tamer (2021). The graph at the top of the figure has the track end at vT=0, rather than $v_T=v_B$. The result of this error is seen most obviously in **Fig. 8c**; the contours indicate that the along-track rate is less than the bulk rate, which contradicts the Ketcham and Tamer model.*

True; Figure 4 shows the corrected model in red and the original one in Figure 8 in black. It also indicates the data range on which the model is based except for the 10 s measurement, which is unconvincing in my opinion. There is indeed something amiss with the left tip of the track in Figure 8c, which I will of course correct.

**Figure 4.** Original (black) and corrected (red) linear $v_T$-model used for calculating the progress of the etchant along an unannealed confined track in Durango apatite (5.5 M HNO$_3$ at 21°C). The shaded areas indicate the database for the vT- model, excluding the 10 s (wrong?) and 15 s (not used) mean track lengths (TK20 Table 3).

[Figure]

This error should however not distract from the significance of Figure 8, which is undiminished. That is to illustrate that a linear $v_T$-model produces tracks that are never observed, either as confined tracks or as surface tracks. The confined tracks can of course be disappeared by a nifty selection criterion; that is the advantage of models. The point is however that it must also agree with the experience of trackers **at the microscopes.**

*[line 450-451] "... except for perhaps 1 µm at either end" seems to admit that **track etch rate does vary**, as the thinning of the track toward its tip is impossible to miss. Furthermore, although Fig. 8 has mathematical problems, it's notable that the only place where the variation in etch rate is similarly impossible to miss is in the last µm or so before each tip.*

Indeed, "$v_T$" decreases, or the track would be infinite. How this comes about is not clear to me. I propose that there is a first phase of staggered etching (JT17) which causes narrowing and rounding, and a later phase when the tips are terminated by faces with the lowest etching apatite etch rates ($v_R$; AS21). Depending on whether one considers individual tracks or the average of a population: this can be interpreted as a decreasing $v_T$ (TC19, TK20, KT21), a transitional rate $v_L$ (Laslett et al., 1984; AS21, JA22) or, based on single-track step-etch data (JT17), erratic length increments of individual tracks due to closely spaced "gaps". This seems to have some support from electron microscopic observations (Paul and Fitzgerald, 1992; Paul, 1993). One can discuss forever whether real tracks look anything like their predicted shapes in Figure 8.

I have a related question: in TK23 polygonal track tips appear shortly after ~10 s "effective etch time". How were the effective etch times determined and how does this square with a linear $v_T$ model and isotropic $v_B$ (TC19; TK20; KT21)? Is this not as flagrant a contradiction between model and observation as Figure 8?

*[line 468-469] Is such curvature unobserved, or just **unobservable**? What curvature there is predicted by the model along the midsection of the track is extremely subtle, and arguably beyond the resolution of optical microscopy, with its diffuse track edges…*

"Unobserved", in the sense that in five decades of confined track measurements, no one has reported a shape like in Figure 8. I have several thousand images of confined tracks (AS21; JS22) but not one like in Figure 8, not even when the apatite etch rate perpendicular to the track is more than twice the assumed isotropic value (AS21).

How am I supposed to respond to an assertion that "there is something there, but you cannot see it"?

*[line 478-480] … for example, the white outlines in Fig 9a,b are sometimes on the inside edge of the blurred region, sometimes on the outside edge. One could draw a curved line on the left side of Fig 9b that is a **scaled version of 8c**.*

No.

*[line 480] **Excess** compared to what? I note that in Fig 9e there are several shorter, under-etched tracks at the edge of visibility, some marked with white arrows and some without. What baseline is the author comparing to?*

This refers to: "*A linear $v_T$ model creates an excess of underetched tracks, e.g., whenever etching starts at the end of a track or its effective etch time is less than the immersion time.*". "Excess" refers to the >80% of **modelled** tracks that are judged to be underetched, compared to those deemed acceptable for measurement. I gather from the $v_B/v_T$-condition that they are "observable", but excluded; have these tracks been "observed"?

*[line 490-492] A bizarre but clarifying assertion. In the Ketcham and Tamer (2021) model, $v_T$ **approaches** $v_B$ at the ends, not zero. Only in the author's attempt to reproduce it does $v_T$ approach 0.*

I admit the mathematical error and will correct it (Figure 3). I must however take back two silent concessions that I made: (1) the access time to α is 6.7 s (instead of 6.0 s); (2) the cross-over from α to β takes 1.3 s (instead of 0 s). The length of β in Figure A3 should be 15.4 μm; the correct value was used in the calculations.

*[line 554] **Accelerated length reduction** was posed as (and still is) an intentionally generic, non-interpretive term that encompasses gap formation but leaves open the option for other possibilities.*

I accept that that is the intended meaning. However I think it is time to come down from the fence and let go of "**other possibilities**". "*I am sure there are still plenty of mistakes in the theory I will offer here, and I hope they are bold ones, for then they will provoke better answers by others*", Daniel Dennet (1983). Prophetic words!

*[line 591-592] And yet such an increase is very clearly present in Tamer's data (10s experiments in Tamer and Ketcham 2020b, Fig. 1). It's not terribly scientific to simply **ignore the data** the contradicts one's conclusion. Can the author provide at least a hypothesis for the incompatibility between the step etch data and those presented here? Which data are more reliable, confined length or inferences from circles on blurry outlines?*

I am confused. I thought I was the one with the **data** (2000+ single-track lengths, widths, angles, …), and TK20 and KT21 were the ones with the **model**, suspended on four to seven mean track lengths (not even length distributions, let alone single track lengths; forget the angles), mostly related to the final 1 μm length increase.

I repeat that the 10 s mean fossil track length (9.11±0.3 μm; TK20; KT21) is inconsistent with an earlier measurement by Murat Tamer (12.5±0.2 μm), as it is with other measurements, including mine (Figure 2). This also applies to the induced tracks where the one measurement that has been confirmed (CA20) is discarded.

*[line 593-595] It's not clear what an "excess" of confined tracks has to do with the projected lengths of **sur­face-intersecting tracks**, the vast majority of which start etching immediately.*

A linear $v_T$ model must produce a massive excess of tracks with lengths between zero and maximum, that are never seen. As far as the confined tracks are concerned, they can be "disappeared" using an *ad hoc* operator bias or tip criterion (KT21 Figure 8). Who can protest? But that does not work for surface tracks, such as ones between Figure 8c and d, which are short and pointed, even after a full immersion time. These should produce an excess of tracks with short projected and etchable lengths, but instead there is a deficit.

*[Figure 10, Figure A3, line 510] These figures also show the author's **mathematical error**. Calculated track etch times are much too long for the linear model, because the author assumes a $v_T$ of zero at the track tip, rather than $v_B$. There is no explanation for how 6s was determined or estimated to be the time required to start etching the first track. And, I have to say, the track segment next to the alpha sure looks like it has curved walls to me, and not just in the last micrometer…*

I corrected the mathematical error and did a quick calculation. I am afraid that the news is not good for the KT21 model. But leaving the numbers aside: how serious is a $v_T$ model that needs to be "saved" (or not) by appealing to a bulk etch rate $v_B$ (0.022 μm/s) which exceeds $v_T$ over 0.11 μm at each end of the track. The 6.0 s access time is calculated from the maximum width of α and the apatite etch rate perpen­dicular to it. This gives its effective etch time, which, subtracted from the immersion time, gives the access time (6.7 s).

The track next to α is a dipping surface track; I leave it to the experts to evaluate the significance of its "curvature".

*It's very worth noting that this is a beautiful **track-in-track-in-track** (TINTINT) image, the first I'm aware of in the literature, and poses an excellent test for the Ketcham and Tamer (2021) variable along-track etching model – can all that etching occur in the time given? I've attached a spreadsheet that demonstrates that it can, though not if it really took 6 seconds to start the first etch. If one drops that time to **2 seconds**, the model predicts the position of all track tips to **within 1 s** of the end of etching at 20s. One can also increase the start time a bit by making some other assumptions that are within the measurements. It also bears mentioning that Tamer and Ketcham (2020) may not have measured a track such as beta, because the top tip is very indistinct. In any event, it would be good to know the basis of that 6s determination, short of which I'll assume for now that my model passed this test.*

What in heaven does it prove that one can (almost) obtain the desired result by changing the input data? What difference does it make if TK20 would have measured β or not? It is there and it is etched; that is what matters.

I have a more serious suggestion: let's for the sake of discussion assume that the maximum etchable length ($L_{lat}$) of the fully etched track β (experts can examine the image stack) is not the default 17 μm, but 16 μm. This alone raises the time to reach its furthest endpoint by almost 10 s, showing how unstable the model is.

*[Appendix A] Here the author has attempted to derive a simpler set of equations than those listed by Ketcham and Tamer (2021) for their model. There could well be a more elegant formulation than mine (to paraphrase Dutch grandmaster J. van der Weil, "I am a butcher, not an artist"), but these are not yet it owing to assuming that $v_T$ = 0 at the tip. One obvious place things are wrong is the case marked (1) va = c a , which should be va = c a + $v_B$, and similarly (2) should be va = c(l/2 – a) + $v_B$. How the author calculated c, though not spelled out, was probably also a bit off (should be (vmax – $v_B$)/(l/2)).*

I changed the *c*-value from 0.200 to 0.197 and calculated, as before, with v = max($v_T$, $v_B$), although the latter should have no effect on Figure A3, as $v_B > v_T$ over the last 0.11 μm of the 17 μm latent tracks (Figure

4)? Appendix A would then correctly describe the $v_T$ model, as such, rather than an inapplicable ($v_B$, $v_T$) model?

*[Figure A2] The distances between the dots are not consistent with each other; if one measures them in pixels, there seems to be about a vertical **distortion** of about 5%, in both Figures A2 and 10, assuming the markers correspond. The measurements are provided on another tab in the attached sheet. It's not clear if this was because the picture was subtly and inadvertently scaled unevenly at some point before or when it was pasted into this document; since both images are distorted by a similar amount, I'd guess it was at some earlier stage in the process.*

I have no knowledge of a distortion of Figure A2 or 10. I captured the images with the same microscope and camera as the other tracks in this work and measured them with the same technique as Murat Tamer used for the others.

I have measured confined tracks for more than 35 years, without, I hope, producing too much nonsense. I believe therefore that I may appeal to colleagues familiar with my work to vouch for me that I can measure tracks.

R. Jonckheere
Freiberg, 26.07.23

---

## Author Comment (AC2)

**On etching, selection and measurement of confined fission tracks in apatite**

RC1: *anonymous reviewer comments, August 1, 2023*

AC2: replies by Raymond Jonckheere, August 1, 2023

*This is a very **interesting** and **important** paper, in principle suitable for publication in Geochronology. The author examined the confined fission tracks (FTs) in apatite in detail, with a particular focus on the shape and width of etched tracks. He analyzed four samples of Durango apatite annealed at different temperatures, and measured geometrical parameters (length, width, angle to **c**-axis, etc.) of horizontal confined FTs by utilizing induced FTs in pre-annealed apatite. This is an **original** investigation with **unique** geometrical analysis of confined FTs at different stages of annealing, and thus has an important **impact** on the FT thermochronology.*

However gratified I am by the reviewer's praise, it would be more convincing if he did not fall over himself to **reject** our manuscript at the end of one page of vacuous comments, without having troubled to read the actual content.

***However**, I found following important issues/pitfalls in the present manuscript that should be treated appropriately before publication:*

*(1) First of all, there is no description about the assessment of uncertainty of individual data in the geometrical analysis, particularly the width of FTs, which is likely the key parameter for reliable confined FT length analysis. I agree with the author's point of view that the assessment of track width is the key, but then, the author should explicitly describe in the text the uncertainty (i.e., **accuracy** and **precision**) of track width measurement.*

This is not a genuine comment; I cannot think of a substantial fission-track publication ever in which this has been done.

As suggested before, I am prepared to upload images of all the confined tracks with the measurements as shown in Figure 1.

*(The author merely gives in Table 1A an "Error" of **0.01 micrometre** (= 10 nm) which is amazingly small for optical microscopic observation.) Otherwise, it may result in the overinterpretation of the obtained data within the range of uncertainty, and lead to total misunderstanding of the phenomena. Note that this issue involves the **propagation** of analytical **errors** in calculating model parameters, such as track etch rate.*

In Table 1A, the first 0.01-µm error is on the mean of $r_0$ (mean: 0.69 µm; standard deviation: 0.23 µm). A 1-2% relative error on a mean of 629 measurements is normal. The calculation can be checked in the supplement.

Furthermore, this has nothing whatsoever to do with the precision of a single microscope measurement.

On reflection, columns 6 to 10 of Tables 1A and 1B present no useful information; I therefore propose to delete them.

The comment is disingenuous; I am not digressing on textbook error propagation for equations as trivial as (1)-(6).

*(2) The **assumption** should be more explicitly documented for calculating the etching rate (and other parameters) from the observed geometrical information (i.e., length and width of a part of confined FTs). The documentation needs to be given in the relevant part of the text, not only giving a series of equations. Otherwise, it may be difficult for readers to follow the logic of the study.*

I am confident that the logic of our manuscript is evident to someone who has read it. The equations are explained in Aslanian et al. (2021; eq. 2-6), and numerical examples are discussed in detail in Jonckheere et al. (2022).

*For example, the author gives the variation of effective etch time versus angle to **c**-axis (Fig. 6), calculated from **track etch rate** values (Fig.7; **constant etch velocity** is assumed without explicit documentation). Then later in the text, he discusses the validity of assuming constant etch velocity throughout an **entire track length** (Fig. 8).*

The effective etch time of a track is calculated from its width and the apatite etch rate, **not** the track etch rate $v_T$ (eq. 1a).

The assumption of a constant track etch rate underlies all practical calculations of cone angles and etching efficiencies since Fleischer and Price (1963). E.g., Fleischer et al. (1975), Tagami and O'Sullivan (2005), Hurford (2019). The discussion in our manuscript, prompted by Tamer and Ketcham (2020) and Ketcham and Tamer (2021), shows that it is right. Had it not been so, our work would have come to a different conclusion.

We nowhere assume a constant etch rate over the **entire** track length, but only over the straight sections we measured.

*Such a framework of the paper is just confusing and I suggest **reorganizing** the text in a more appropriate logical flow. Concerned with this, the author should better document/discriminate between physical theory, experimental observation, model calculation, and interpretation. These appear to be confused/contaminated from each other in places in the current text. This makes it difficult to **understand** the significance of new findings in the study.*

I find that reading a manuscript helps to understand it. I am not reorganizing it based on such a vague comment.

*(3) We see many typos of the experimental parameters in the text and figures that are similar to each other. This makes it further difficult to read the paper correctly.*

Like all conscientious authors, I will check the manuscript for typos and consistent use of definitions and symbols.

*Because of these issues, I judge that it is not appropriate to accept the paper at its present form of data presentation and interpretation. Therefore, I regret to suggest **rejecting** the paper, with a strong encouragement for the author to resubmit the material as a more carefully **reorganized** and **rewritten** manuscript.*

It is obvious that this is not an objective review, but an attempt to inflict the greatest possible damage. On both occasions when the reviewer refers to actual data, he shows a complete lack of understanding of the subject.

Apart from that, not one comment refers to the content of the manuscript except for what can be gleaned from the abstract and a glance at Table 1. The reviewer must realize that this is not acceptable reviewing practice.

The random hand-waving about error propagation and structural reorganization are tried and tested tools in the arsenal of unscrupulous "anonymous" reviewers intent of causing maximum disruption with least effort.

Sincerely,

Raymond Jonckheere
Freiberg, august 1st 2023.

---

## Author Comment (AC3)

**On etching, selection and measurement of confined fission tracks in apatite**

RC2: *comments by Raymond Donelick, August 1, 2023*

AC3: replies by Raymond Jonckheere, August 2, 2023

**General Comments**

*It is an honor and pleasure to review this excellent paper. Well done Dr Jonckheere! You have quantified the mental picture in my head of the many nuances of confined fission track lengths in apatite. I highly **recommend** this paper be accepted for publication and I leave it to the author to choose to make the minor clarifications I request below. I have not read the comments that have been posted for this paper (as of 31 July 2023) and may not do so.*

I consider it a privilege to have my manuscript reviewed by Dr. Donelick. This has nothing to do with his favourable comments. More than one recent submission has been the target of malicious reviews, which have nothing to do with the scientific content (review #1). I receive no reward and get no promotion for peer-reviewed papers. Publishing has, in my experience, become a disagreeable and exhausting process, and there are easier, perhaps fairer, ways to put out data or ideas for the next generations to run with, ignore, or dismiss.

I am shocked at some of the manuscripts submitted for publication, and reviews that seem like there is an all-out war raging.

However, this review restores my faith in the scientific process. **It is worth it** when someone of the stature and unmatched competence of Raymond Donelick takes the time to have a civilized discussion about one's scientific work.

**Specific Comments**

*Introduction: The biases discussed do not include mention of other decisions facing an analyst such as: 1) Is that a naturally etched fission track? Sample TI of Carlson et al. (1999) exhibits many naturally etched fission tracks near natural grain surfaces. 2) Is that feature a fission track at all or some outlier that should be removed from the dataset? Like potentially naturally etched fission tracks, some potentially non-fission track features can etch like fission tracks but be too long (I use 19 microns as a cutoff for low-temperature natural and laboratory samples). Or perhaps an etched feature looks like an etched fission track but it is too short for the current fission track population(s) under study. Human analysts are smart enough to make these decisions and these decisions should be done openly.*

I did not know about the Tioga apatite, but I have also observed naturally etched tracks in some samples. The present work is however carried out on induced tracks Durango apatite, and, with at most a handful of exceptions, all the confined tracks are TinT's. Therefore I did not want to further complicate this manuscript. I also intend to put all the tracks and all the measurements on a suitable server. Carolin Aslanian is also re-measuring the KTB with the intention of creating a virtual KTB (isothermal holding?) profile, showing each measured confined track, allowing everyone to select and measure the tracks according to their insight.

*Lines 46-50: I can never produce such a plot as discussed here. The notion of measuring sufficiently-etched fission tracks (apples) and contaminating those data by also measuring under-etched fission tracks (oranges) is not appealing to me. My plots would instead have no data (shorter etch times), then very little data, then sufficient data (longer etch times). I do not mix apples and oranges when it comes to measuring confined fission tracks in apatite.*

I understand. I once measured confined tracks etched for 10 s (5.5M $HNO_3$; 21 °C) with the idea to get at information about the track structure that is erased at longer etch times. Step-etch data show however that the etchant progresses micrometres further along an underetched track than can be seen with a microscope. That makes such measurements meaningless, except perhaps in terms of observation biases. I believe that from the beginning the common-sense approach has been to etch the tracks to their ends, or slightly beyond, and to wrest what information we can from collections of tracks etched to their **intrinsic full lengths**.

*Equations: It is these equations that can be used to show that non-elliptical, polar coordinate plots (length, angle to c-axis) of horizontal confined fission track lengths from a single population represent a mixture of apples (sufficiently etched tracks defining the ellipse, usually at higher angles to c-axis) and oranges (under-etched tracks falling short of the ellipse, usually at lower angles to c-axis).*

I know those plots, and I admit that our group has produced some as well. We propose to define an effective etch time window to weed out underetched and overetched tracks, and are considering other measures as well.

*Lines 172-177: Thank you for reproducing my work and that of Dr. Bill Carlson. The agreement between our works is truly independent and it is not accidental, but instead demonstrates that we as analysts can behave like machines with the ability to maintain biases within ourselves and share those biases with other machines.*

I hesitated to discuss the lengths as it had been done before. I now believe that it is perhaps the most important outcome of this work, in particular in a climate where each measurement is considered to be the result of one bias or another, and therefore meaningless. The fact that three investigators separated by two decades and two oceans agree on a substantial set of measurements without conferring is indeed **not accidental**.

*Lines 184-187: My first submission of Donelick (1991) was to EPSL and a reviewer rejected the paper on the basis that "they [polar-coordinate plots of fission track lengths] are not ellipses", because his/her published mixtures of apples and oranges did not match my apples-only data.*

I am curious who, in 1991, had both the data or the confidence to disagree with the elliptical model, but I can make a guess.

*Lines 199-209: Excellent discussion of this interesting issue. As we are after the mean and the mean ranges vary accordingly, it seems it should be possible to prove mathematically this is precisely how it should be.*

I am grateful for the support. I am also conscious that, not without genuine hesitation, I am trespassing on your own work. But I think it was worth commenting if it helps someone to better understand it or to make use of it.

*Lines 248-261: In Donelick et al. (1999) I proposed a surface energy model for fission track annealing in apatite. In this model, converging track sides (during annealing) parallel to **c**-axis are flat and perpendicular to **c**-axis are rough with pyramidal faces. In this model, at high angles to **c**-axis and high degrees of annealing you get the results of Donelick et al (1999) – systematic accelerated length reductions - with the occasional Green et al. (1986) – segmentation where one or more opposing and 'rogue' pyramids intersect – along with the tips pinching off and appearing to diffuse into the bulk crystal as in Paul and Fitzgerald (1992). Surface energy minimization explains anisotropy and all experimental observations.*

*Lines 270-271: This statement is totally consistent with the surface energy minimization model above.*

I agree that the surface energies control the track boundaries during annealing, and, I would add, during track formation and etching. I'd need convincing, however, that the walls of tracks perpendicular *c* are made up of pyramidal faces. I see how this is suggested by the terminations of apatite crystals, but why do we not observe a pyramidal texture when etching basal faces? I guess that these ideas go back to Nichols and Mullins (1965)? I found them too difficult to follow but I would be interested to discuss them sometime.

I believe we are all agreed on unetchable gaps at the level of our observations. I commented on accelerated length reduction because it is confusing that both terms have existed next to each other for so long. If they do not mean the same, accelerated length reduction must be more than, or different from, gap formation. If it is different and continuous (no gaps) then is it anisotropic length reduction? Or is it something else?

*Lines 346-348: I hazard to say that only analysts willing to mix apples and oranges have this problem as your (and my) experiments demonstrate. This is especially true for samples with low confined fission track densities such as highly annealed experiments or the many, many, many low-fission-track-density natural samples studied without using $^{252}Cf$/particle accelerator ion implantation.*

I wish we hadn't measured half the low-density samples that we did. It not only tempts the analyst to measure underetched tracks but also to count and measure tracks in faces that are far from prismatic. As Carolin Aslanian will show at the next conference, we combine deep ion implantation with etching for 40

s (Ito, 2004); this produces numbers of confined tracks in common grains of the same magnitude as the that of the surface tracks; we then use our width measurements to select those within a predefined etch time window.

*Line 424: I see this minor but significant effect for $l_{a0}$ and $l_{c0}$ and a correlation between these and Dpar and Dper, respectively. I presented these data in Amsterdam (2004) but do not have the reference.*

It makes me wonder why I haven't plotted the ellipse axes against effective etch time. It is a great idea; it would be the right graph for seeing if there is a change of the anisotropy during etching. Likely not, but worth checking.

The Amsterdam reference is: O'Sullivan P.B., Donelick R.A., Ketcham R.A. (2004) Etching conditions and fitting ellipses: what constitutes a proper apatite fission-track annealing calibration measurement? 10th International Fission Track dating and Thermochronology (Conference), 8-13 August 2004, Amsterdam, DVL-10-O.

*Lines 461-463: Track shapes are due to surface energy ratios, the surfaces being different apatite crystal planes in contact with the etchant of some molarity and at some temperature. For a given apatite, we expect surface energy ratios to change – thus changing track shape - with changing acid strength and temperature (and possibly pressure).*

Doubtless. I think that at low enough concentrations diffusion rates (stirring) begin to have an effect as well.

*Lines 510-511: I have observed huge differences among unidirectional, low angle-of-incidence $^{252}$Cf tracks in apatite. The differences are much more than those observed for confined fission tracks from $^{235}$U or $^{238}$U but keep in mind that confined tracks include the other half of the track that is missing when using $^{252}$Cf. It seems likely that $^{252}$Cf length correlates with nucleus energy.*

I have never worked with a $^{252}$Cf source but with accelerator ions; I cannot say that I observed such an effect.

*Lines 544-546: Thank you for showing this. Thank you also for making available your imagery for posterity. In the current state-of-the-art, measurements for fission track experiments should be performed using such imagery and such imagery should be made available to anyone freely who requests it. The days of "just trust me" should end.*

This was not the aim of this investigation but I believe it is perhaps the most important result. A colleague wrote in relation to confined track length measurements: "*It all depends on how we look at things and not how they are in themselves (Jung, 1941)*". I cannot think of more depressing (woke) principle in science. It is important to prove that we **cannot** measure whatever we want. I believe our work has reconfirmed that there is a truth, and that it is accessible and meaningful. Why else would we bother? And it does not take much to find it; to also end with a quote: "*There is one thing even more vital to science than intelligent methods, and that is the sincere desire to find out the truth, whatever it may be*" (Charles Sanders Peirce).

**Clarifications Requested**

*Lines 193-195: Do you mean the standard deviations of **c**-axis projected fission track lengths? Please clarify.*

Yes; I will check if the symbol ($s_{PM}$) is consistent with its use elsewhere in the text and in the Figures, and explain.

*Lines 237-239: I assume you mean "...shorter projected mean lengths" "...longer projected mean lengths" "...order of projected mean lengths". Please clarify.*

Yes; I will correct it.

*Lines 241-242: It might be useful to cite here Jensen and Hansen (2021) and related comments https://gchron.copernicus.org/preprints/gchron-2021-8/#discussion.*

I can add the reference although I must admit that I did not understand much of that paper or the discussion.

*Lines 427-421: It is worth noting here that I measured the Carlson et al. (1999) data using transmitted light only*

Yes. I think this proves that conscientiousness is the mark of the scientist, i.e. attempting to measure an unbiased, representative sample, and resisting the temptation to go for the low hanging fruit (picking the fat tracks that light up in reflected light) or a shortcut to a quick result (measuring nice tracks in non-prism faces).

Thanks for taking the time to review,

Raymond Jonckheere
Freiberg, august 2nd 2023.

---

## Author Comment (AC4)

Geochron 2023-13

**On etching, selection and measurement of confined fission tracks in apatite**

CC2: *comments. by Richard Ketcham and Murat Tamer, August 9, 2023*

AC4: replies by Raymond Jonckheere, August 11, 2023 and Final response

**Background**

Some time ago Murat Tamer told me that he had no Microscope. I sent him the images I had made for this investigation. I explained how we measure and calculate. He did this fast and well but no substantial interpretation or manuscript was forthcoming. I took over and had a manuscript towards the spring. We had an agreement that Murat Tamer would have two go's at submitting it as corresponding author, after which I would. In April I declined a request from Chemical Geology to review the manuscript "*How many vs. which ...*", by Tamer and Ketcham, despite serious misgivings about the manner in which our work was presented.

I posted the current manuscript on ResearchGate to make clear that I would not allow it to be suppressed, which I had reason to suspect. This elicited immediate criticism from Richard Ketcham. I offered to submit it to Geochronology, which had not been my initial choice, in order to allow him to comment, since he could not review it, because of his collaboration with Murat Tamer and some results that contradicted his KT21 model. I also advised Murat Tamer, who was still on board at that time, that Richard Ketcham should not be reckless in his comments. First of June, I submitted the current manuscript, which was posted on the 16th. I had made some changes to the calculation in favour of the KT21 model in order not to make it look too bad. Nevertheless, Murat Tamer still asked the associate editor not to be named as co-author. There was never, before or since the slightest scientific argument that might have prompted this decision.

Richard Ketcham posted a first critical comment on the 13th. I thereupon contacted associate editor Shigeru Sueoka to ask if he could withdraw this comment, or if I could be excused from replying. I did that because I knew that it would leave me no alternative than to demolish the KT21 model, which would cause embarrassment and friction. I was told that I had to respond "in order to stimulate discussion", which I did on the 26th.

I was informed on July 29th that the discussion was extended to August 9th because "additional" referee comments were needed. On August 1st an anonymous reviewer and Raymond Donelick posted their reviews, to which I replied on the 3rd. On the 9th, less than five hours before the deadline, Richard Ketcham and Murat Tamer posted a joint second comment in response to my original replies, to which I now have to answer. On the 11th these comments have been removed and I am asked to "finalize" before the 18th. What do I do? Is it unreasonable to ask if there has been private communication or agreement between Murat Tamer, Richard Ketcham and associate editor Shigeru Sueoka that influences the handling of this manuscript?

The extended deadline has expired; there is an insubstantial review and a favourable review by Raymond Donelick, but no "additional" review. Instead Richard Ketcham and Murat Tamer have appointed themselves reviewers instead of commentators, offering to allow my manuscript to be published against certain concessions, and the recommendation that one of them should be co-author. The associate editor is someone with whom Murat Tamer has an ongoing collaboration. Although I have no part in that, I assisted the irradiation of their samples, for which I decided to also bear the expense after Murat Tamer had withdrawn as co-author.

**General replies**

For discussion, it is perhaps useful to again explain the disagreement with TK20 and KT21. (1) The manner of our $v_T$-measurements assumes that $v_T$ *is* **constant** along most of the length of a fission track. If it isn't, and $v_T$ varies along a track, as the linear model requires, then our data are meaningless. (2) In contrast to TK20 and KT21, our samples provide no evidence for an **increase** of the track etch rate $v_T$ following partial annealing.

As far as I can tell, the latest comments are restatements of the earlier ones, further insinuations about preconceived notions, and deft use of words taken out of context, but there is nothing substantial related to any facts.

This goes to show that my essential moderate criticism, not so much of the KT21 model, as of the manner that it has been used to undermine my manuscript, has not come across as I hoped. Permit me therefore to be somewhat blunt: in the words of Wolfgang Pauli, the KT20 and KT21 model is "*not even wrong*", it is absurd.

Suppose I want to **determine** the variation of $v_T$ along a fission track ($v_T$ profile). I can proceed as follows (Figure 1a): digitize the track contour, which gives me a function $w(x)$, calculating the derivative $|dw/dx|$ then gives me:

[Figure]

$$\left| \frac{dx}{dw} \right| = \left[ \frac{1}{v_B} \right] v_T(x)$$

The result is an actual, continuous $v_T$-profile along a section of the track. Doing that for a larger number of tracks permits to investigate not just the $v_T$- variation along a track, but also from track to track and the correlation with their lengths, orientations, etc. That is how to do it, and how we learn that the tracks are straight.

One can also make a model, based on the assumption that all tracks have the same length and the same $v_T$-profile, regardless of their formation, orientation or individual annealing histories (fossil tracks); we add an isotropic $v_B$, and an unheard-off operator bias and roundedness selection criterion. Then we forget the entire midsection of the tracks, including the notion of width, and measure their lengths instead (Figure 1b). This construct is fitted to a handful of step-etch data; if I may be blunt, in my opinion, half are bulk-etching and the other half invisible tracks. Nevertheless it also gives a solution within the pre-defined constraints. Thus understood that is alright, and why I recommended publication of the KT21 manuscript.

But it is beyond a shadow of a doubt that such a model carries **zero weight** when it comes to real $v_T$-data. And I draw the line when it excuses questioning the competence and integrity of those who disagree with it. Endless tiresome tirades about unclear images, microscope resolutions, repeat measurements, assertions, assumptions, preconceptions ... indicate commentators scared out of their wits. It is undignified and presumptuous to think that one can browbeat lesser colleagues into concessions by threatening them with such a little stick.

Wherever the model contradicts even a single measurement of a single track, the model loses. **With respect to the corrections**, this entitles me to dismiss all model implications as **irrelevant**. Despite their bitterness, the commentators might consider my motivation to not have included the above comments in the manuscript.

**Specific replies**

**p. 1**: "*This accommodation is not sufficient to match the KT21 model ...*". The equations and calculations are correct, except for one adjustment, i.e., the slope $c$ should be $(1.7-0.022)/(17/2) = 0.197$, instead of $1.7/(17/2) = 0.200$.

**p. 2**: "*... the data be made available.*" The data are available to Murat Tamer; I will take advice from the journal about it.

Murat Tamer is welcome to the fossil track images. I leave it to his discretion to acknowledge their origin, and if relevant, that of the concept of the experiment. I suggest to integrate them with the

comprehensive fossil track data that we sent him before, and to publish the results together with Leila Sarkosh, who did the measurements. On the other hand, I do not wish to be involved in the publication.

**p. 3** : There is a long section related to the independent 10 s (15 s) length data contradicting those in TK20 and KT21.

"*... finding tracks at the edge of visibility ...*": how can measurements subject to extreme selection bias form a valid basis for a $v_T$-profile, which is a fixed physical property of the tracks (given mineral and etchant)?

"*the JB03 data ... are for apatite from the Fish Canyon Tuff ...*": I thought this was about a $v_T$-model, but (apart from the etchant) it look as if it is a "*Durango-Tamer-2020*" $v_T$-model; what would the 2013 model be like?

**p. 4** "*Figure 2 in RJ's reply is a modified version of Figure 9 in TK20.*" I changed colour to greyscale to do the commentators a favour; the human eye is much more sensitive to shades of grey than to colour, in particular red.

"*... we don't believe ...*", "*... we also don't believe ...*"; those are not arguments about facts that require an answer.

"*... presuming ill intent ("political and calculating").*" I will forego several karma points here and stick to the rule: *tell the truth or at least don't lie* (Jordan Peterson). I supervised Murat's Bachelor and Master theses and spent countless hours talking to him since. I did my utmost to dissuade him from retiring as co-author. I can be mistaken, but his motivation is obvious to me, in one short sentence: "Richard Ketcham is a much more promising vehicle for future publications than Raymond Jonckheere". He is right, it is true, as this manuscript proves again. And it is a valid reason; I am therefore not suggesting "*ill intent*". It was a sensible and acceptable career choice. But there can be no suggestion at all of a scientific motivation. That said, since the danger appears less than he feared, he is welcome to come on board again, assuming he can make his mind up if he wants to sink this ship or sail in it.

**p. 5** "*The author is confusing his feuds ... all scientific reasoning is always improved by considering applicable, independent work*". Yeah ... this comes from scientists who twice published a sensational $v_T$-increase due to annealing based on five measurements, without verification or reference to published and other accessible data, and a model that has never been – and will never be – tested against any data at all.

**p. 6** "*... it seems that Equation 3b should use an arctan, not an arcsin ...*". The arcsin is correct; Murat Tamer can explain.

**Conclusions**

**Reviewer #1** (anonymous): no corrections.

**Reviewer #2** (Raymond Donelick): all suggested corrections accepted.

**Comments #1 and #2** (Richard Ketcham and Murat Tamer): I can find nothing in these comments that indicates an error in the manuscript; all the data are valid and the equations and calculations are correct, except for a minor (<2%) adjustment of one of the constants used. I will of course make this correction.

As to the co-authorship of Murat Tamer, I would be pleased to have him on board again. However, I make no concessions in return regarding the content of this manuscript. I urge him to accept, as I consider the present criticism a mere token opposition to safe face. It will serve him better to be seen to abandon the TK20-KT21 models, as he started to do in TK23, and fall in line with the work of our group.

R. Jonckheere
Freiberg, August 11